TOPICAL REVIEW

# Voluntary activation of muscle in humans: does serotonergic neuromodulation matter?

Justin J. Kavanagh[1] 🆔 and Janet L. Taylor[2,3] 🆔

[1] *Neural Control of Movement laboratory, Menzies Health Institute Queensland, Griffith University, Gold Coast, Australia*
[2] *Centre for Human Performance, School of Medical and Health Sciences, Edith Cowan University, Perth, Australia*
[3] *Neuroscience Research Australia, Sydney, Australia*

Handling Editors: Laura Bennet & Jean-Claude Béïque

The peer review history is available in the Supporting Information section of this article (https://doi.org/10.1113/JP282565#support-information-section).

**Abstract** Ionotropic inputs to motoneurones have the capacity to depolarise and hyperpolarise the motoneurone, whereas neuromodulatory inputs control the state of excitability of the motoneurone. Intracellular recordings of motoneurones from *in vitro* and *in situ* animal pre-

---

**Associate Professor Justin Kavanagh** leads the Neural Control of Movement laboratory in the Menzies Health Institute Queensland (Griffith University). His research team examines mechanisms that contribute to muscle activation in humans, where his work has a strong emphasis towards using drug interventions to explore how neurotransmitters contribute to muscle activation and fatigue.

---

parations have provided extraordinary insight into the mechanisms that underpin how neuro-modulators regulate neuronal excitability. However, far fewer studies have attempted to translate the findings from cellular and molecular studies into a human model. In this review, we focus on the role that serotonin (5-HT) plays in muscle activation in humans. 5-HT is a potent regulator of neuronal firing rates, which can influence the force that can be generated by muscles during voluntary contractions. We firstly outline structural and functional characteristics of the serotonergic system, and then describe how motoneurone discharge can be facilitated and suppressed depending on the 5-HT receptor subtype that is activated. We then provide a narrative on how 5-HT effects can influence voluntary activation during muscle contractions in humans, and detail how 5-HT may be a mediator of exercise-induced fatigue that arises from the central nervous system.

(Received 31 March 2022; accepted after revision 12 July 2022; first published online 21 July 2022)

**Corresponding author** J. Kavanagh: Neural Control of Movement laboratory, Menzies Health Institute Queensland, Griffith University, Gold Coast, Australia. Email: j.kavanagh@griffith.edu.au

**Abstract figure legend** Inputs to neuromodulatory receptors on motoneurones, such as those involved in the serotonergic system, modify the motoneurones' responsiveness to ionotropic input. The release of serotonin (5-HT) into the spinal cord is linked to the level of motor activity being performed, where 5-HT can increase the discharge rate of motoneurones via excitatory 5-HT receptors on the soma and dendrites. This in turn can lead to increased voluntary muscle activation (VA) and maximal force generation. However, intense release of 5-HT onto motoneurones may lead to a spillover of 5-HT into extracellular compartments to activate inhibitory 5-HT receptors on the axon initial segment. This can cause a reduction in motoneurone discharge rate, thus decreasing VA and maximal force generation. To gain insight into the serotonergic contributions to muscle activation in humans, pharmacological interventions have been employed to enhance the concentration of 5-HT in the central nervous system or activate selective 5-HT receptors.

## Introduction

The final common pathway in the motor system is the motor unit, which consists of the $\alpha$-motoneurone in the spinal cord and all muscle fibres that it innervates. Given that the motor unit links the central nervous system to skeletal muscle, efficient and purposeful movement can only be achieved by tightly controlled motoneurone activity. The firing pattern of a motoneurone is produced by converting synaptic inputs into action potentials, where the regulation of input to output is determined by the intrinsic electrical properties of motoneurones (Kernell, 2006). Hence, to gain a better understanding of how the nervous system regulates motor activity, it is of great importance to explore how inputs to the motoneurone influence activation of muscle.

Synaptic inputs to motoneurones will typically activate two classes of receptors: ionotropic and neuromodulatory. In a general sense, inputs to ionotropic receptors are command signals from descending motor pathways, peripheral afferents, or spinal interneurons, which have the capacity to depolarise and hyperpolarise the motoneurone. Typically, ionotropic receptors are ligand-gated ion channels which directly gate ion flow into cells to generate either an excitatory or inhibitory response in postsynaptic neurons (Alexander et al., 2011; Eccles & McGeer, 1979; Heckman & Binder, 1991; Powers et al.,

2002). In contrast, inputs to neuromodulatory receptors control the state of excitability of the motoneurone through intracellular signalling pathways that modify the motoneurone's responsiveness to ionotropic input (Eccles & McGeer, 1979; Elliott & Wallis, 1992; Fedirchuk & Dai, 2004; Heckman, Hyngstrom et al., 2008; Murray et al., 2011). The role that neuromodulators play in activating motoneurones has received considerable attention in the past 30 years. Most notably, intracellular recordings of motoneurones from *in vitro* and *in situ* animal preparations have provided considerable insight into how electrical properties of motoneurones are influenced by neuromodulators. However, fewer studies have attempted to translate findings from intracellular experiments into humans, as there are considerable challenges in reproducing reductionist experiments in an intact human neuromuscular system.

There are several neuromodulator systems that regulate the excitability of cortical and spinal motor circuits, and many excellent reviews have been written that describe the effects of each system on motor activity (e.g. the cholinergic system: Deffains & Bergman, 2015; Jones, 2008; Mille et al., 2021; Naicker et al., 2017; the dopaminergic system: Arber & Costa, 2022; Ikeda et al., 2015; Klaus et al., 2019; Sharples et al., 2014; and the noradrenergic system: Balaban, 2002; Benarroch, 2018;

Fung et al., 1994). Our Topical Review will provide a unique summary regarding how serotonergic neuromodulation contributes to voluntary activation of muscle in humans. To achieve this, the functional anatomy of serotonergic pathways with regard to the motor system will be outlined before summarising human experiments that assess the role of this neuromodulator in performing unfatigued and fatiguing muscle contractions.

## The serotonergic system

Serotonin (or 5-hydroxytryptamine, 5-HT) is a mono-amine neurotransmitter that modulates the intrinsic properties of neurons in the central nervous system (CNS). In the periphery, it is synthesised in enteric mucosal cells, influences gut motility among other functions, and is stored in platelets (Terry & Margolis, 2017). However, 5-HT is unable to cross the blood–brain barrier and must be synthesised within the CNS from its precursor, tryptophan. Hence, the primary site of CNS 5-HT synthesis in vertebrates is the raphe nuclei of the brainstem (Hery et al., 1982; Hornung, 2003; Imai et al., 1986; Pollak Dorocic et al., 2014). At a cellular level, 5-HT is released into synapses and exerts pre- and postsynaptic effects when it binds to membrane receptors. It is often suggested that the termination of 5-HT's effects at the synapse occurs with reuptake into presynaptic terminals or glial cells via monoamine transporters. However, serotonergic effects on neuronal function may not necessarily end with reuptake of 5-HT, as activation of G-protein-coupled receptors can elicit long-lasting changes in ion channel function (Pavlos & Friedman, 2017). Seven classes of 5-HT receptors have been identified (Lucas & Hen, 1995). They are all G-protein-coupled receptors (Hoyer & Martin, 1997) except for the 5-HT$_3$, receptor which is a ligand-activated channel (Thompson & Lummis, 2006). However, the principal 5-HT receptors expressed by motoneurones are 5-HT$_{1A}$, 5-HT$_{1B}$, 5-HT$_{2A}$, 5-HT$_{2B}$ and 5-HT$_{2C}$. (For comprehensive reviews on 5-HT receptor subtypes in the CNS refer to Barnes & Sharp, 1999; Nichols & Nichols, 2008; Peroutka, 1990; Perrier et al., 2013.)

From the brainstem, neurons from a group of serotonergic nuclei *ascend* to distribute across many areas of the brain, including the frontal lobe where 5-HT can be released in the cerebral cortex (Fig. 1*A*). A high density of 5-HT fibres has been identified in motor areas of the rat and primate cortex (Vertes, 1991; Wilson & Molliver, 1991), but the role of these rostral raphe projections in motor activity has not been well-defined. Cortical structures predominantly express receptors from the inhibitory 5-HT$_1$ family and to a lesser extent the excitatory 5-HT$_2$ family (Celada et al., 2013; Pazos & Palacios, 1985; Pazos et al., 1985), where

5-HT$_{1A}$ is expressed extensively in laminae II/III of the motor cortex and 5-HT$_{2A}$ receptors are uniformly distributed across motor cortex laminae (Joyce et al., 1993; Vitrac & Benoit-Marand, 2017). It appears that depleting brain 5-HT in rats has the effect of reducing the excitability of neural circuits in motor cortex (Scullion et al., 2013), and administration of selective serotonin reuptake inhibitors (SSRIs) enhances excitability in motor cortex circuits in humans (Gerdelat-Mas et al., 2005; Ilic et al., 2002). Thus, a likely key role of 5-HT in the motor cortex is modulation of facilitatory intracortical motor circuits. 5-HT also appears to play a critical role in plasticity of motor circuits, as SSRI administration combined with either non-invasive brain stimulation

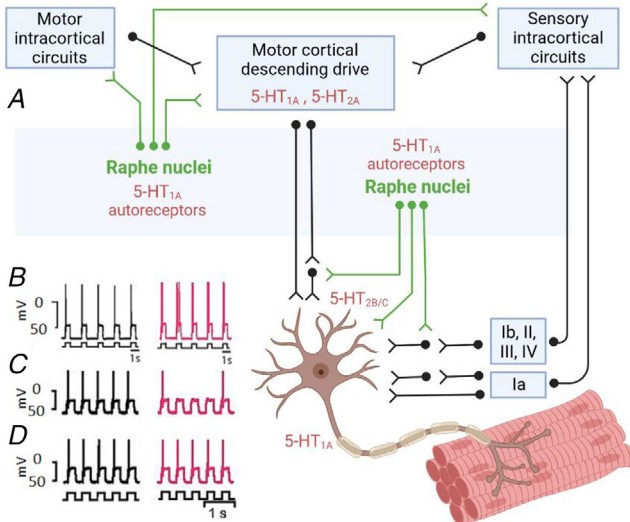

**Figure 1. Serotonergic activity modulates cortical and spinal motoneurone activity**
*A*, serotonin synthesis in the mammalian CNS occurs in the raphe complex of the brainstem. Rostral raphe nuclei ascend primarily to the forebrain to innervate virtually all regions. Caudal raphe nuclei project mainly to the cerebellum and spinal cord, where the monosynaptic raphe–spinal pathway releases 5-HT on the soma and dendrites of spinal motoneurones. Ionotropic inputs from descending motor cortical pathways, peripheral afferents, or spinal interneurons, have the capacity to depolarise and hyperpolarise the motoneurone. However, activation of 5-HT receptors, predominantly in the 5-HT$_1$ and 5-HT$_2$ families, regulates the state of excitability of the motoneurone via intracellular signalling pathways to modify the motoneurone's responsiveness to ionotropic input. *B*, intracellular recordings from motoneurones in an adult turtle spinal cord preparation (black). The number of action potentials evoked by depolarizing current pulses injected in the soma is increased following 1 s of 40 Hz electrical stimulation of the dorsolateral funiculus (red), which contains descending serotonergic fibres of the raphe–spinal pathway. *C*, lengthening the duration to 3 min of 40 Hz electrical stimulation inhibited motoneurone firing compared to the control, which is attributed to 5-HT spillover onto extracellular 5-HT$_{1A}$ receptors on the axon initial segment. *D*, 3 min of 40 Hz stimulation in the presence of the 5-HT$_{1A}$ antagonist WAY-100635 (20 $\mu$M) removed the inhibitory influence of 5-HT. Data sourced from Cotel et al. (2013).

or motor practice can influence excitatory synaptic plasticity. In particular, there is evidence to suggest enhanced long-term potentiation-like plasticity from paired associative stimulation (Batsikadze et al., 2013; Nitsche et al., 2009), whereas changes in plasticity from practicing motor tasks has been observed in most (Classen et al., 1998; Loubinoux et al., 2002; Pleger et al., 2004), but not all (McDonnell et al., 2018), experiments that have assessed cortical plasticity after SSRI ingestion. Functional magnetic resonance imaging has identified several inter-connected structures that are affected by enhanced 5-HT concentration, where short-term drug-induced neuro-plastic changes are associated with enhanced activation of primary motor and premotor cortices, as well as posterior supplementary motor areas (i.e. executive motor areas of the cortex) (Loubinoux et al., 1999).

Most of the work that has examined 5-HT effects on motor cortical activity has employed magnetic or electrical stimulation techniques to explore cortical plasticity, or neuroimaging techniques to map changes in regional activity due to changes in neurotransmitter concentration. These approaches provide immense insight to the cortex itself, but do not necessarily link 5-HT changes in the CNS to motor performance. Instead, considerably more is known about how 5-HT release on motor circuits in the spinal cord directly affects motor activity. From the brainstem, the raphe–spinal pathway *descends* to the spinal cord to form well-defined synapses with motoneurones, afferent neurons, and interneurons (Fig. 1*A*) (Alvarez et al., 1998; Kawashima, 2018; Ridet et al., 1994). 5-HT release into the dorsal horn and inter-mediate zone of the spinal cord can cause remarkably complex outcomes at the neuronal level, as 5-HT can either depress or facilitate transmission in afferent fibres (Belcher et al., 1978; Jordan et al., 1979; Todd & Millar, 1983), and the response to 5-HT can differ depending on the threshold for afferent activation (Belcher et al., 1978; Jankowska et al., 1993). Adding further complexity to our understanding of how 5-HT regulates sensory neurones in the spinal cord is that 5-HT's effects may be exerted presynaptically and postsynaptically. In particular, modulation by 5-HT of the actions of muscle spindle (Ia fibres) and tendon organ (II fibres) afferents on spinal interneurons depends on the type of afferent that is activated and the functional type of the interneuron (Jankowska et al., 2000). Thus, it is difficult to predict how motor function may be influenced by the actions of 5-HT on afferent neurons and interneurons. Instead, the most direct effects of 5-HT on motor function are via the monosynaptic connections to the motoneurones from the fibres in the raphe–spinal pathway to the motoneurones. There are approximately 1500 serotoninergic contacts on each motoneurone (Alvarez et al., 1998), where the release of 5-HT occurs 0.5–1.0 s following the onset of detectable neural activation of locomotor muscles (Noga

et al., 2017). The intensity of 5-HT release from the raphe-spinal pathway, but not the ascending pathway, is thought to correspond to the intensity of motor activity being performed by upper limb (forelimb) and lower limb (hindlimb) muscles. This viewpoint has predominantly been based on cat studies that show activity in the raphe nuclei is abolished during sleep (Cespuglio et al., 1981; McGinty & Harper, 1976), but near-linearly scaled to locomotion speed (Jacobs & Fornal, 1997; Jacobs et al., 2002; Veasey et al., 1995). Incremental increases in treadmill walking speed correspond to incremental increases in the discharge rate of single raphe-spinal fibres (Veasey et al., 1995), which potentially points towards a contraction intensity dependent mechanism of 5-HT release. On face value, this suggests a very useful motor control strategy, in which strong contractions cause large release of 5-HT to amplify signals from synaptic inputs. However, a phenomenon exists where the presence of too much 5-HT in the CNS can limit the ability of motoneurons to activate muscle.

## Dual role of 5-HT on motoneurone activity

There is widespread agreement that 5-HT promotes depolarisation in motoneurones, which arises from several concomitant mechanisms. In particular, cellular and molecular experiments have identified that 5-HT can facilitate a rectifying inward current (Hsiao et al., 1997; Takahashi & Berger, 1990), facilitate low threshold $Ca^+$ currents (Berger & Takahashi, 1990), or inhibit $K^+$ leak conductance (Elliott & Wallis, 1992; Perrier et al., 2003) to increase neuronal excitability. 5-HT can also increase discharge rate via mechanisms that reduce the amplitude of slow and medium afterhyperpolarisation phases that follow the action potential (Bayliss et al., 1995; Grunnet et al., 2004; Hounsgaard & Kiehn, 1989; Hounsgaard et al., 1988; Wikstrom et al., 1995). Although 5-HT can reduce hyperpolarisation by decreasing the threshold for generating $Na^+$-based action potentials (Fedirchuk & Dai, 2004), 5-HT also generates a $Ca^+$-dependent plateau potential by reducing a $K^+$ current responsible for slow afterhyperpolarisation (Hounsgaard & Kiehn, 1989). In general, the functional role of 5-HT on motoneurone activity has been revealed by slice preparations, partially intact animal preparations, and simulations. With these approaches, it has been possible to examine the interaction between physiological 5-HT concentrations and 5-HT receptor subtypes. Electrical stimulation of the dorso-lateral funiculus (DLF; contains descending serotonergic fibres) in integrated preparations of the adult turtle spinal cord has been shown to promote the release of end-ogenous 5-HT into the ventral horn of the spinal cord (Delgado-Lezama et al., 1997; Perrier & Cotel, 2008b; Perrier & Delgado-Lezama, 2005). In this circumstance,

motoneurone recordings have confirmed that discharge rate increases with brief bursts of 5-HT release (Fig. 1*B*) (Hounsgaard & Kiehn, 1989; Perrier & Delgado-Lezama, 2005; Perrier & Hounsgaard, 2003), and this increase is mediated by activation of 5-HT$_{2B/C}$ receptors located on the dendrites and the cell body of the motoneurone (Cotel et al., 2013; Jackson & White, 1990).

In addition to 5-HT exerting excitatory effects on motoneurones, the presence of this neuromodulator has also been observed to create an inhibitory effect on motor activity. Early *in vivo* work that recorded motoneurone activity in cats observed that 5-HT injected in the vicinity of the motoneurone caused hyperpolarisation of the resting membrane, which led to a failure of the motoneurone to fire (Phillis et al., 1968). An underlying mechanism of 5-HT-mediated inhibition of motoneurone activity was clarified in a series of elegant experiments by Cotel and Perrier. Using preparations from the turtle spinal cord, it was revealed that the ability of motoneurones to fire is inhibited when 5-HT is iontophoretically applied close to motoneurone soma (Perrier & Hounsgaard, 2003) and when prolonged DLF stimulation causes large endogenous release of 5-HT onto motoneurones (Fig. 1*C*) (Cotel et al., 2013; Perrier et al., 2017). The mechanism underlying 5-HT's dual excitatory and inhibitory effects aligns with the compartmentalisation of 5-HT receptor subtypes expressed on motoneurones. The monosynaptic connections that raphe-spinal neurons have with motoneurones causes direct 5-HT release onto receptors in somatodendritic regions to enhance motoneurone excitability. However, activation of 5-HT$_{1A}$ receptors expressed in perisomatic regions inhibits Na$^+$ channels responsible for the genesis of action potentials, which has the effect of preventing motoneurone firing (Perrier & Cotel, 2008a). Given that the inhibitory 5-HT$_{1A}$ receptors are only located on the axon initial segment, and the axon initial segment is devoid of serotonergic innervation, activation of 5-HT$_{1A}$ receptors can only occur if extracellular concentrations of 5-HT are large enough to spill over onto the axon initial segment (Cotel et al., 2013; Perrier & Cotel, 2015; Perrier et al., 2017). These findings provide a potential cellular mechanism for central fatigue and provide a foundation to investigate if 5-HT activity during prolonged bouts of physical activity reduces motor output in humans. However, it is worth noting that these findings are only reported for adult turtle motoneurones and may not be applicable for other species. Indeed, a recent study has identified 5-HT boutons present on the axon initial segment of rodent motoneurones (Deardorff et al., 2021), and compartmentalisation of 5-HT receptor subtypes on human motoneurones is yet to be detailed. Finally, other activity-dependent mechanisms may be influenced by serotonin. For example, sodium–potassium pump

activity can cause inhibitory effects on motoneurone discharge during rhythmic activity (Zhang & Sillar, 2012; Zhang et al., 2015; Picton & Sillar, 2016). In tadpoles, this effect can be modulated bidirectionally through 5-HT$_{2A}$ and 5-HT$_7$ receptors (Hachoumi et al., 2022), although this has not yet been demonstrated in mammals.

## Is serotonergic neuromodulation reflected in voluntary muscle activation in humans

Although it is difficult to identify how individual neuromodulators contribute to motor activity, simulations indicate that the net effect of maximal neuromodulation may be a three- to five-fold amplification of the currents that the motoneurons receive from synaptic inputs (Hultborn et al., 2003; Lee & Heckman, 2000). Hence, the availability of 5-HT is critical to enhancing the gain of motoneurone output, and ultimately, activating the muscle to produce force. Data from healthy adults performing maximal effort contractions mostly support this finding. However, the magnitude of response in human experiments is small (Fig. 2*A* and *B*). Following the administration of the SSRI paroxetine to increase serotonin availability, the force of maximal voluntary contractions (MVC) has been noted to increase 1.5–4.5% in several experiments employing elbow flexion protocols. Similarly, voluntary activation increased by 0.3% when

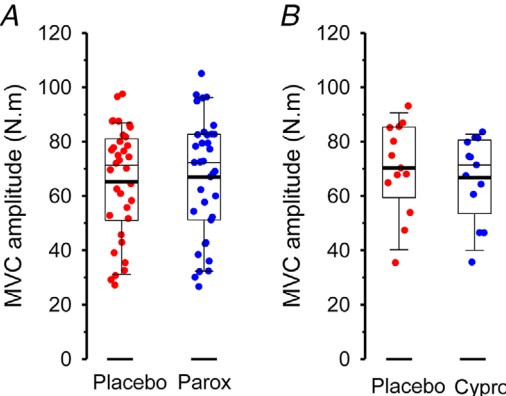

**Figure 2. Elbow flexion MVC torque after ingestion of paroxetine or cyproheptadine**
*A*, the SSRI paroxetine (Parox) exerts small, but significant, increases in torque during brief unfatigued MVCs (paired Student's *t*-test, *P* = 0.046). Given that voluntary activation of biceps is high in healthy individuals (e.g. 97–99%), a ceiling effect exists whereby only small increases in voluntary activation and elbow flexion torque are possible. *B*, competitive antagonism of the 5-HT$_2$ receptor via cyproheptadine (Cypro) causes significant reductions in the ability to generate elbow flexion torque during brief unfatigued MVCs (paired *t*-test, *P* = 0.003). Data are sourced from a combination of studies from the host laboratory using healthy adults aged 20–30 year (Henderson et al., 2022; Kavanagh et al., 2019; Thorstensen et al., 2020; Thorstensen et al., 2021).

measured using cortical stimulation (Thorstensen et al., 2020) and by ∼1.3% when measured using motor nerve stimulation (Kavanagh et al., 2019; Thorstensen et al., 2020). A finding of only small drug-related increases in voluntary activation is perhaps not surprising, as increases in motoneurone excitation during near-maximal contraction intensities produce only small changes in interpolated twitch amplitude (Herbert & Gandevia, 1999). Hence, voluntary activation of the biceps (which is calculated from interpolated twitches) may be as high as 98% or 99% in healthy individuals, and any intervention that may increase activation is limited by a very close ceiling (i.e. 100% voluntary activation). Nevertheless, an enhanced ability to activate the muscle is consistent with 5-HT effects that are mediated by excitatory somato-dendritic 5-HT$_2$ receptors on motoneurones. Consistent with this, the administration of a 5-HT$_2$ competitive antagonist leads to the opposite effect on maximal force generation. Maximal elbow flexion force declined by ∼3% after the administration of cyproheptadine (Thorstensen et al., 2021). Antagonism of the 5-HT$_2$ receptor also leads to a reduction in rate of force development when performing rapid dorsiflexions, which is mediated by reduced firing rates in motor units of the tibialis anterior (Goodlich et al., 2022). Thus, enhancing 5-HT availability during strong unfatigued contractions increases force generation, whereas blocking 5-HT$_2$ receptor activity decreases force generation via neural mechanisms in healthy individuals.

Persistent inward currents (PIC) play a critical role in setting the gain of motoneurones. PICs are mediated by voltage-gated sodium channels (Na$_v$1.1 and Na$_v$1.6) and L-type voltage-gated calcium channels (Ca$_v$1.2 and Ca$_v$1.3) on somato-dendritic surfaces of motoneurones (Heckmann et al., 2005; Schwindt & Crill, 1977; Schwindt & Crill, 1980). In mammals, there are believed to be equal contributions from slow activating L-type Ca$^{2+}$ current and a fast activating persistent Na$^+$ current to the PIC (Heckman, Hyngstrom et al., 2008). However, recent evidence shows that non-linearities in motoneurone discharge may have more complex mechanisms including activation of TRPm5 and inactivation of Kv1.2 channels (Bos et al., 2018; Bos et al., 2021). Nonetheless, PICs are strongly influenced by monoamines. Activation of PICs causes a remarkable amplification of depolarising drive to the motoneurone and evokes a strong acceleration of motoneurone firing rate (Bennett et al., 1998; Hounsgaard et al., 1988; Lee & Heckman, 1998). Thus, it might be expected that PICs may be partially responsible for 5-HT-mediated increases in voluntary activation during MVCs. In humans, the amplitude of PIC activation can only be estimated. The accepted technique for this estimation for an individual motoneurone compares the estimated synaptic input that recruits a motoneurone to that at its derecruitment (i.e. the motoneurone's hysteresis). PICs provide little drive to the motoneurone at recruitment as initiation of an action potential and the activation of PICs occur at a similar membrane potential (but see Afsharipour et al., 2020 for important nuances with regard to variations in PIC threshold). By comparison, PICs are fully active after the motoneurone has been firing for ∼0.5–1.0 s (Heckman, Johnson et al., 2008; Lee & Heckman, 1996). Thus, at derecruitment, PICs contribute part of the depolarising current for motoneurone firing so that less synaptic input is required.

The paired motor unit technique uses the firing rate of low threshold motor units, which already have PICs fully active, as estimates of synaptic input to other motor units in the same pool. The measure that is calculated is the difference in frequency of firing of the lower threshold motor unit at recruitment and derecruitment of the higher threshold motor unit (known as $\Delta F$). Discharge rates are recorded from intramuscular electromyography electrodes (Foley & Kalmar, 2019; Gorassini et al., 2002; Udina et al., 2010) or extracted from high-density electromyography arrays placed over the muscle (Afsharipour et al., 2020; Hassan et al., 2020). The idea that the influence of serotonin on human motor performance is mediated via PICs is supported by pharmacological studies in a small number of healthy participants ($n = 3$), in whom $\Delta F$ increased with ingestion of an SSRI or decreased with a 5-HT$_2$ receptor antagonist (D'Amico et al., 2013). Because $\Delta F$ has only been calculated during submaximal ramp-shaped contractions, it is unknown if PICs are directly linked to enhanced MVCs in humans. However, it is known that PICs associated with gastrocnemius medialis increase from 10% to 20% MVC and PICs associated with soleus increase from 10% to 30% MVC during slow ramped plantarflexions (Orssatto, Mackay et al., 2021). These findings provide support that PIC activation in humans is linked to voluntary drive and hence 5-HT release. The difference between gastrocnemius and soleus also highlights that PICs, and their influence on motoneuron intrinsic properties, differ between muscles and motoneuron subtypes (Huh et al., 2017; Lee & Heckman, 1998). Reductions in motoneurone PIC may also provide a mechanism for loss of force production from skeletal muscle. In younger populations, there is evidence that $\Delta F$ reduces up to 25% for soleus motoneurones following passive stretching of the plantarflexors, which may explain some of the reductions in force associated with stretching protocols (Trajano et al., 2020). In older populations, $\Delta F$ is lower than in young adults for soleus and tibialis anterior (Orssatto, Borg et al., 2021), and biceps and triceps brachii (Hassan et al., 2021), and may contribute to age-related reductions in motoneurone firing rates and muscle force. However, the mechanisms underlying the reductions in PIC activity are not known. Conceivably, monoamine release onto the motoneurones could be reduced or the

motoneuronal response to monoamines could be altered by changes in receptors. Alternatively, PICs are sensitive to inhibitory synaptic input and could be turned off by added inhibition.

## Is prolonged release of serotonin a mechanism of fatigue in humans?

Serotonin has long been implicated in the development of fatigue in humans, where the performance of prolonged exercise increases blood tryptophan levels and thus 5-HT synthesis in the CNS (Newsholme et al., 1987). The 'central fatigue hypothesis' was introduced in the 1980s, when it was proposed that enhanced concentration of brain 5-HT induced negative effects on arousal and mood, and increased lethargy and sleepiness. It was postulated that this mechanism could influence the perception of effort and, therefore, fatigue (Newsholme & Blomstrand, 2006; Newsholme et al., 1987). Following this original hypothesis, several studies used pharmacological interventions to modify CNS 5-HT levels during prolonged cycling, and found that cycling performance was sometimes (Meeusen et al., 2001; Roelands et al., 2009; Strachan et al., 2004), but not always (Strüder et al., 1998; Teixeira-Coelho et al., 2014; Wilson & Maughan, 1992), limited by enhanced 5-HT concentrations. These mixed findings are most likely a reflection on the complexity of neurotransmitter systems, and no single neurotransmitter is responsible for exercise-induced central fatigue. Indeed, pharmacological interventions that antagonise D2 dopaminergic receptors (Thorstensen et al., 2018) and inhibit noradrenaline reuptake (Klass et al., 2012; Klass et al., 2016) have been shown to reduce voluntary muscle activation during fatiguing contractions in humans. Another difficulty with interpretation of pharmacological interventions during whole-body exercise is that widespread, and often contradictory, interactions of serotonin occur for the motor system. Even at a supraspinal level, if increased cortical 5-HT does reduce motivation and increase effort through actions at high level motor areas, increased motor cortex excitability with SSRI ingestion suggests that corticospinal pathways may be facilitated (Gerdelat-Mas et al., 2005; Ilic et al., 2002).

Benchtop experiments cannot replicate the same physiological conditions associated with exercise-induced fatigue. However, recordings obtained from animal preparations strongly suggest that activation of 5-HT$_{1A}$ receptors in the spinal cord would limit the ability for the CNS to activate muscle, and thus contribute to central fatigue. This specific mechanism appears relevant to humans, as ingestion of the 5-HT$_{1A}$ receptor agonist buspirone suppresses spinal motoneurone excitability (D'Amico et al., 2017) and reduces the capacity to perform prolonged bouts of exercise in healthy individuals

(Marvin et al., 1997). However, exogenous activation of the 5-HT$_{1A}$ receptor may cause different responses to naturally occurring 5-HT dynamics, so we performed several experiments to assess muscle activation when endogenously released 5-HT is accumulated in the CNS. Fatiguing prolonged repeated *maximal* effort contractions were performed with and without enhanced availability of 5-HT caused by ingestion of an SSRI. We observed that MVC torque and time-to-task failure were reduced when more 5-HT was available (Fig. 3*A*) (Kavanagh et al., 2019). That is, more fatigue developed more quickly. It was evident that this decline in performance had neural origins, as fatigue-related failure of voluntary activation was greater in the presence of higher 5-HT concentrations (Fig. 3*B*). Additional experiments were performed using the hand muscle abductor digiti minimi, where fatigue-related reductions in F-wave persistence and F-wave area following a prolonged MVC were substantially greater with greater 5-HT availability. As F waves are a marker of motoneurone excitability, these findings indicated that reduced excitability of spinal motoneurones contributed to the added central fatigue caused by 5-HT reuptake inhibition. Consistent with the idea of serotonin spillover as described above, the SSRI-related accumulation of 5-HT in the spinal cord could be due to strong serotonergic drive occurring when neural drive to the target muscle was high. The high levels of synaptic serotonin could then spill over to activate the extra-synaptic inhibitory 5-HT$_{1A}$ receptors on motoneurones to suppress voluntary muscle activation.

Prolonged low-intensity contractions can also lead to substantial declines in motor performance due to central fatigue (Sogaard et al., 2006). During the performance of a 15% MVC for 30 min, the development of muscle fatigue was demonstrated by a reduction in force of occasional brief MVCs and central fatigue by associated declines in voluntary activation. By comparing performance with and without serotonin reuptake inhibition, we were able to test if 5-HT effects progressively limit muscle activation over time as the sustained contraction presumably causes a sustained release of 5-HT in the CNS (Thorstensen et al., 2020). Contrary to our hypothesis, muscle force, voluntary activation and corticospinal excitability were not affected by 5-HT reuptake inhibition during the prolonged low-intensity contraction (Fig. 3*C* and *D*). The absence of 5-HT effects may have occurred due to two possible mechanisms. First, during prolonged low-intensity contraction, neural drive to the muscle may not have been of a sufficient intensity to cause intense release of 5-HT from the raphe-spinal pathway onto motoneurons. Hence, serotonin reuptake may have kept up with serotonin release despite the SSRI. Second, serotonergic drive may have declined throughout the contraction and caused a reduction of 5-HT release onto motoneurones. The latter mechanism has support from

cat experiments as the firing of raphe-spinal neurons is reported to progressively decrease during sustained physical activity. In single raphe-spinal neurons activity decreased by up to 50% after 40 min of treadmill walking (Fornal et al., 2006). Thus, serotonergic neuromodulation may not be an endless resource during muscle contractions. As yet, it is not clear what level and duration of voluntary contraction is enough to elicit fatigue-related motoneurone inhibition through serotonin spillover. Our experiments suggest that 30 min of a sustained 15% MVC is insufficient, but 40–60 s of a sustained MVC is sufficient.

## Afferent feedback can potentially regulate 5-HT effects at the motoneurone

Voluntary activation of muscle can be modulated by muscle afferent feedback. This presents a challenge for understanding motoneurone excitability, as ionotropic input received by the motoneuron comprises concurrent excitatory (descending drive and muscle spindle afferents) and inhibitory (group Ib, III and IV) synaptic activity

that can be modulated by 5-HT (Fig. 1). Adding to this complexity, most muscle afferents synapse on interneurons in the spinal cord to cause excitation and/or inhibition in multiple target muscles. Thus, examining the action of individual afferents, and recording activity from individual muscles, may not reflect the net output of the motor system (D'Amico et al., 2014).

Although the amplitude of dendritic PICs is directly proportional to the intensity of brainstem neuromodulatory drive, inhibition from low-threshold sensory inputs has a strong suppressive effect on PIC amplitude. When inhibition is strong enough, the PICs created by the monoaminergic drive are reduced or even deactivated (Hultborn et al., 2003; Kuo et al., 2003). *In vivo* cat experiments have revealed that reciprocal inhibition may be especially critical for regulating PIC amplitude during functional motor activity. Even minor rotations of the ankle joint that cause barely detectable changes in reciprocal inhibition of motoneurones will reduce PICs by ~50% (Hyngstrom et al., 2007). The likely candidate for this afferent mechanism is Ia disynaptic reciprocal

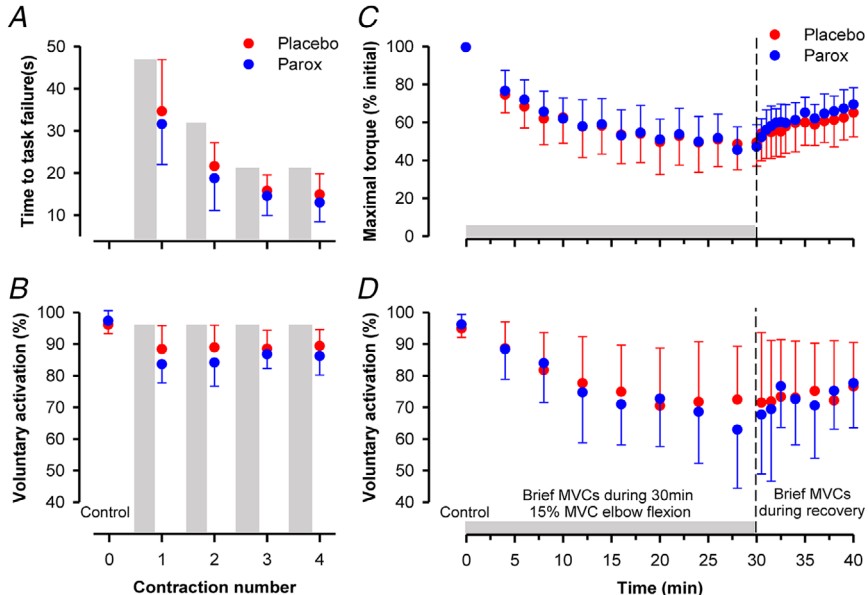

**Figure 3. The effect of enhanced availability of 5-HT on maximal and submaximal fatiguing isometric contractions**
Paroxetine (Parox) was used to enhance 5-HT concentrations by reuptake inhibition. Motor nerve stimulation was used to assess voluntary activation of the elbow flexors. *A*, time-to-task failure was assessed for four sustained maximal elbow flexions, where each contraction was maintained until force declined to 60% MVC. Grey bars indicate a fatiguing contraction has occurred. A rest period of 40 s occurred between maximal contractions. *B*, each fatiguing contraction was followed 3 s later by a resting twitch, and then 3 s later by a brief maximal contraction with superimposed twitch, where voluntary activation was calculated as: [1 − (superimposed twitch/resting twitch)] × 100. Enhanced 5-HT concentration caused a significantly greater fatigue-related reduction in time-to-task failure ($P = 0.038$) and voluntary activation ($P = 0.048$). *C*, elbow flexion torque has also been assessed during a sustained 15% MVC, where the contraction was held for 30 min and fatigue responses continued to be monitored for 10 min of recovery. *D*, superimposed and resting twitches were obtained periodically throughout the contraction protocol and during recovery to calculate voluntary activation. There were no changes to fatigue-related reductions in MVC torque ($P = 0.376$) and voluntary activation ($P = 0.505$) due to enhanced 5-HT availability. Data sourced from Kavanagh et al. (2019) and Thorstensen et al. (2020).

inhibition evoked by length changes in the antagonist muscle, as other afferents are relatively insensitive to the changes in muscle length that occurred with passive rotation of the cat ankle in the experiment (Hyngstrom et al., 2007). The notion that reciprocal inhibition can dampen the effects of PICs has also been demonstrated in humans, where 128 Hz vibration of the tibialis anterior tendon to activate dorsiflexor muscle spindle afferents decreases $\Delta F$ for both the soleus and medial gastrocnemius during 30% MVC plantarflexions, and vibration of the Achilles tendon decreases $\Delta F$ for the tibialis anterior during 30% MVC dorsiflexions (Pearcey et al., 2020). Another human study found that 1 Hz electrical stimulation of the common peroneal nerve also has the capacity to reduce $\Delta F$ in medial gastrocnemius motor units in healthy individuals, which builds further support that Ia reciprocal inhibition reduces the contribution of PICs to MU firing in humans. (Mesquita et al., 2022). Interestingly, reciprocal inhibition may solve a motor control problem that arises from the diffuse projection of monoaminergic fibres in the spinal cord. Widespread projections from the serotonergic system will release 5-HT on more than one motor pool during voluntary contractions. Hence, 5-HT may simultaneously enhance PIC activity on multiple motoneurones, which has the potential to create a bias towards co-contraction due to increasing excitability in antagonist motor pools (Heckman, Hyngstrom et al., 2008). A reasonable proposal suggests that Ia reciprocal inhibition may promote voluntary activation during a variety of motor tasks by only enhancing activity in task specific muscles from a background of diffuse excitatory neuromodulation (Heckman, Hyngstrom et al., 2008; Hyngstrom et al., 2007; Hyngstrom et al., 2008). Thus, it is possible that a 5-HT mechanism involving inhibitory spinal circuitry can regulate the amplitude of agonist and antagonist muscle contractions.

## So does serotonergic neuromodulation matter in humans?

Animal experiments provide clear evidence that 5-HT is a potent modulator of spinal circuits and motoneurone output. However, the effects of 5-HT on voluntary muscle activity in humans are less clear. Effects on motor performance during whole-body exercise are inconsistent. This is not to say that serotonergic neuromodulation does not matter for humans, but instead highlights the challenges associated with studying how a complex neuromodulatory system acts during muscle contractions. Controlled experiments using single-joint, single muscle, contraction protocols have found that maximal force can be changed by altering 5-HT activity in the CNS. The differences observed in voluntary activation are small but are present despite an otherwise

intact system and the actions of other neuromodulators, including noradrenaline. This suggests that serotonin has a non-redundant role in maximal voluntary contractions but still begs the question of exactly how important it is in typical motor tasks. Nonetheless, 5-HT-related changes in muscle activation typically emerge with strong contractions for both the unfatigued and fatigued motor system. Thus, it appears that the magnitude of descending drive to the muscle may be aligned with the level of 5-HT neuromodulation in humans. Indeed, we are beginning to reveal evidence where 5-HT's effects may be scaled to the intensity of muscle activation in humans (Goodlich et al., 2022; Henderson et al., 2022).

An inability to quantify 5-HT release onto motoneurones, as well as quantifying the binding affinity to different 5-HT receptor subtypes, presents a challenge for interpreting any neuromodulation experiment (not just human studies). Participant safety is an additional challenge in human experiments that use pharmacology to manipulate neurotransmitter activity. Human studies must operate within a window of safe drug administration, and typically use therapeutic doses of 5-HT-modulating medications. Thus, very little is known about how dosage effects influence 5-HT activity in humans. Pharmacological interventions are also non-specific, and will not have 100% effectiveness when competitively agonising, antagonising or inhibiting 5-HT reuptake in humans. Individual variation in pharmacokinetics and drug responses also poses a challenge when studying humans, but variability in 5-HT responses is also a prevalent feature of *in vitro* and *in vivo* animal experiments that examine motoneurone and afferent activity. It is our position that none of these factors should prevent human 5-HT research from continuing, but instead should encourage scientists to further explore the relationship between neuromodulation and voluntary muscle activation.

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

## Additional information

### Competing interests

None.

### Author contributions

Both authors contributed to the conception and the design of this work, as well as the drafting and final approval of the manuscript. All authors have read and approved the final version of this manuscript and agree to be accountable for all aspects of the work in ensuring that questions related to the accuracy or integrity of any part of the work are appropriately investigated and resolved. All persons designated as authors qualify for authorship, and all those who qualify for authorship are listed.

### Funding

No funding was received for this work.

### Acknowledgements

Open access publishing facilitated by Griffith University, as part of the Wiley – Griffith University agreement via the Council of Australian University Librarians.

### Keywords

fatigue, motoneurones, neuromodulation, persistent inward current, serotonin

## Supporting information

Additional supporting information can be found online in the Supporting Information section at the end of the HTML view of the article. Supporting information files available:

**Peer Review History**

