## [Peer Review History · The Journal of Physiology]

Voluntary activation of muscle in humans: does serotonergic neuromodulation matter?

Justin J Kavanagh and Janet L Taylor
DOI: 10.1113/JP282565

Corresponding author(s): Justin Kavanagh (j.kavanagh@griffith.edu.au)

The following individual(s) involved in review of this submission have agreed to reveal their identity: Simon A Sharples (Referee #1); Gregory EP Pearcey (Referee #2); Tuan V Bui (Referee #3)

Review Timeline:

Submission Date:	31-Mar-2022
Editorial Decision:	07-Jun-2022
Revision Received:	21-Jun-2022
Accepted:	12-Jul-2022

Senior Editor: Laura Bennet

Reviewing Editor: Jean-Claude Béïque

Transaction Report:

Dear Dr Kavanagh, Re: JP-TR-2022-282565 "Voluntary activation of muscle in humans: does serotonergic neuromodulation matter?" by Justin J Kavanagh and Janet L Taylor Thank you for submitting your Topical Review to The Journal of Physiology. It has been assessed by a Reviewing Editor and by 3 expert referees and I am pleased to tell you that it is considered to be acceptable for publication following satisfactory revision. The reports are copied at the end of this email. Please address all of the points and incorporate all requested revisions, or explain in your Response to Referees why a change has not been made. NEW POLICY: In order to improve the transparency of its peer review process The Journal of Physiology publishes online as supporting information the peer review history of all articles accepted for publication. Readers will have access to decision letters, including all Editors' comments and referee reports, for each version of the manuscript and any author responses to peer review comments. Referees can decide whether or not they wish to be named on the peer review history document. I hope you will find the comments helpful and have no difficulty in revising your manuscript within 4 weeks. Your revised manuscript should be submitted online using the links in Author Tasks Link Not Available. This link is to the Corresponding Author's own account, if this will cause any problems when submitting the revised version please contact us. You should upload: - A Word file of the complete text (including any Tables); - An Abstract Figure, (with accompanying Legend in the article file) - Each figure as a separate, high quality, file; - A full Response to Referees; - A copy of the manuscript with the changes highlighted. - Author profile. A short biography (no more than 100 words for one author or 150 words in total for two authors) and a portrait photograph of the two leading authors on the paper. These should be uploaded, clearly labelled, with the manuscript submission. Any standard image format for the photograph is acceptable, but the resolution should be at least 300 dpi and preferably more. You may also upload: - A 'Cover Art' file for consideration as the Issue's cover image; - Appropriate Supporting Information (Video, audio or data set https://jp.msubmit.net/cgi-bin/main.plex?form_type=display_requirements#supp). To create your 'Response to Referees' copy all the reports, including any comments from the Senior and Reviewing Editors into a Word, or similar, file and respond to each point in colour or CAPITALS. Upload this when you submit your revision. I look forward to receiving your revised submission. Yours sincerely, Professor Laura Bennet Senior Editor The Journal of Physiology <https://jp.msubmit.net> <http://jp.physoc.org> The Physiological Society Hodgkin Huxley House 30 Farringdon Lane London, EC1R 3AW UK <http://www.physoc.org> <http://journals.physoc.org> ----- EDITOR COMMENTS Reviewing Editor: This manuscript has been reviewed by 3 expert reviewers. They unanimously enjoyed reading this review, noted its scholarship qualities and ultimately recommend publication. Several minor suggestions and clarifications are proposed and they appear to be expert and reasonable. As such, the authors should have no difficulties in addressing them in a timely manner. ----- REFEREE COMMENTS Referee #1: Kavanagh and Taylor provide a timely review with the goal of highlighting and summarizing recent work in humans that describe roles and underlying mechanisms for 5HT and target receptors in the voluntary activation of muscle in humans. To accomplish this end, they place their human work in the context of multiple animal models that describe how serotonin modulates intrinsic properties and ion channels of motoneurons. The review draws the conclusions that studying 5HT control of muscle activation in humans might not be as straight forward as initially thought and propose/discuss several potential challenges. Because of the cross-human-animal comparison, this review will be of interest and useful to those studying neural control of muscle in humans and likewise, will be of interest to those studying mechanisms governing motoneuron excitability in animal models. This review is well written and was enjoyable to read. That being said, I have several comments that I think should be addressed. It is my hope that these comments will strengthen the review. Comments: Of note, the manuscript was not submitted with line numbers but was generated by the editor upon request. To ensure line number references in this review match that of the document, Page 1 Line 1 starts with 'Journal of Physiology Topical Review', Page 2 Line 31: 'Abstract', Page 2 Line 47: 'Introduction'...etc. Page 2 Line 38-40: the statement saying 5HT can have profound effects on force generation during muscle contractions is a bit of an overstatement as the review highlights recent work that shows a very modest change MVC following manipulation of the serotonergic system. While I can agree that 5HT has a potent effect on firing rates, this doesn't seem to translate to huge changes in maximal muscle force - which is interesting and discussed in the review. I think this discrepancy should either be highlighted in the abstract to provide context for the title of the review. Alternatively, I suggest toning back this statement. Line 73: Why has serotonergic modulation been so extensively studied? While I can agree that it is a potent modulatory system in the control of spinal circuits, it is far from the only one. I think that this would be worth highlighting and would be useful to provide reviews that describe roles for other neuromodulators in the control of spinal circuits/motoneuron excitability (eg. Cholinergic, dopaminergic, noradrenergic, etc.). Line 88: While the effects of ionotropic neurotransmitters may be terminated by removal of the ligand, the effects of neuromodulators, such as 5HT on neuronal function may not necessarily be terminated by reuptake of 5HT given that activation of GPCRs can elicit long lasting changes in ion channel function. This statement should at least be re-written to simply highlight 5HT reuptake, including what is the primary means for 5HT (eg. SERT). Line 125: it is not clear what is meant by evoked responses. Would this be using TMS in humans or in animal models using other approaches? Line 137: worth mentioning is work from Brian Noga's Lab that used voltammetry to show increases in 5HT that occurs on the time scale of seconds during MLR-evoked fictive locomotion in cats (<https://doi.org/10.3389/fncir.2017.00059>). Interestingly, 5HT levels begin to increase prior to the onset of fictive locomotion which is supportive of a role for 5HT in setting the network tone for locomotion. Line 146-150: the three mechanisms described would collectively increase neuronal excitability. I think it is worthwhile using precise language to highlight this here. Eg change modulate on Line 150 to increase. Line 152: It is unclear what is exactly meant by 'inhibition of hyperpolarization'. It is not clear how reducing spike threshold would inhibit hyperpolarization. Lines 152-156 should be rethought and reworded for clarity. Line 160: the DLF is not a specific analogue of the raphe-spinal pathway, but does contain descending serotonergic fibres, in addition to many other. This might be misleading as written. Line 167: which neuron? Line 169: An additional inhibitory and activity dependent mechanism mediated by dynamic sodium potassium ATPase pumps has also been demonstrated by the Sillar lab for control of motoneuron activity in *Xenopus* tadpoles (Zhang et al. 2012; 2015). Importantly, this inhibitory pump is under the control of 5HT receptors (Hachoumi et al., 2022). Similar inhibitory pump dynamics have also been demonstrated in mammalian motoneurons and can be modulated by dopamine

(Picton et al., 2016). These studies are worth highlighting for intrinsic cellular mechanisms that could contribute to the inhibitory (and activity dependent) regulation of motoneuron excitability. Line 172: by 'increased hyperpolarisation' do you mean that the resting membrane potential was hyperpolarized? If so, this should be reworded for clarity. Line 174: while the work was conducted in JF Perrier's lab, I think it is important to acknowledge and the credit should be drawn to the first author, F Cotel, who likely performed the majority of the work. Please rephrase. Line 189: An interesting observation that I think is worth mention, that 5HT fibres have been reported at the AIS of rodent motoneurons (Deardorff, Romer, and Fyffe, 2021), which is in contrast to that has been suggested by Cotel et al. It is possible that this is a species difference. Alternatively, it might be due to differences in motoneuron types, which have not really been explored. Regardless, it still provides a means for direct compartmental modulation of channels that are expressed at the AIS, which differ from those expressed in the dendrites. Line 224: Indeed, previous work has demonstrated contributions of CaV and NaV channels to PIC and PIC-mediated firing properties, and historically it was thought that PICs were relatively straight forward. However more recent work from the Brocard group has suggested that it might be a bit more complex than initially thought as other ion channels can contribute to PIC and PIC-mediated intrinsic properties. Examples include activation of TRPM5 (Bos et al., 2021) and inactivation of KV1.2 (Bos et al., 2018). Further, M-type potassium currents (mediated by KCNQ channels) influence measures of PIC and their effect on intrinsic properties and oppose NaV1.6-mediated inward currents in excitatory interneurons. While this direct interaction has not been shown in motoneurons, it is worth noting given that motoneurons do express KCNQ channels in somatodendritic and AIS compartments alongside NaV1.6 channels (Verneuril et al., 2020). Whilst historically it was thought that PICs were straightforward, it is becoming more apparent that this is not the case as multiple channels contribute to and even oppose their actions in parallel. A similar argument could be made for modulation of 'PIC-channels' as changes in PIC-mediated intrinsic properties could be mediated through changes in some of these other channels. I highly recommend checking these works out and think that these points are worth highlighting in this review. Line 249: I would suggest softening this statement from PICs mediate serotonin's effect on motor performance, to PICs may partially contribute to serotonin's effect on motor performance. As previously highlighted, many other ion channels can be influenced by serotonin. Line 251: PICs and their influence on motoneuron intrinsic properties differ between motoneuron subtypes. This might be a good spot to highlight some of these works (Lee and Heckman 1998, Huh et al. 2017; Sharples and Miles 2021). I think it is also worth highlighting that motoneuron subtypes express different complements of ion channels and possibly (although not known) express different 5HT receptors or have inputs to different regions. Along these lines, studies of different muscles with varying muscle fibre compositions (eg. Soleus, TA, etc) may also respond differently to neuromodulation. These could possibly contribute to task-differences in function and neuromodulatory control. Line 253-256: This statement isn't very clear. I suggest breaking it up a bit. Line 256: Not to mention that changes in the expression of the PIC channels or those that oppose their actions could also contribute. Line 313: suggest toning back language from '...in the spinal cord was due to...' to '...in the spinal cord could be due to...' as although plausible, this is purely speculative. Line 348: Modulated 'by' 5HT. Line 369-373: Maybe not such a recent idea with the citations provided approaching 15 years ago. Suggest rephrasing. Line 375: This paragraph ends kind of open. Is there a possibility to capitalise on new genetic tools in rodents that allow for identification and manipulation of defined interneuron subtypes to advance these ideas further? Are there new methods/technologies/techniques in humans that might provide some insight? Line 377: I'm not entirely sure that this question was answered in this conclusion paragraph. Indeed, challenges are highlighted, but I think it would be worthwhile to outright statement that might provide more of an answer to this question. Line 378: neuromodulation of what? This sentence is rather vague as 5HT is a neuromodulator and it can be inferred that it would contribute to neuromodulation. I would suggest stating that 5HT is a potent modulator of spinal circuits and motoneuron output. Line 393-396: this sentence is a little misleading as it implies that pharmacological manipulation of neurotransmitters in the CNS is unique to humans. However, I think that the main point is detailed in the sentence that follows. Perhaps consider rephrasing these two sentences. Figure 1: I think this figure is a little problematic as it attempts to synthesize data from multiple species (turtle and mammals). It should be made clearer in the figure itself (in addition to caption) where these data are derived otherwise it is a bit misleading. As highlighted above, it should be highlighted that the DLF contains descending serotonergic fibres and is not simply an analogue of the raphe spinal pathway. Referee #2: It was with pleasure that I read the work submitted by Kavanagh and Taylor entitled "Voluntary activation of muscle in humans: does serotonergic neuromodulation matter?" For many years, the field has been left wondering if PICs that are facilitated by 5HT are actually relevant to human function. This topical review starts to address this, and other, aspects of 5HT neuromodulation during normal and fatigued motor output. It is with great certainty that I can say that this article will be highly read and cited. Great work! Below, I have provided some specific comments (with page

; and line [I]) in an attempt to improve the quality of the manuscript: p2; I57: Ionotropic/neuromodulatory is not necessarily a class of input, rather they are inputs that activate classes of receptors. The inputs activate either ionotropic or metabotropic receptors and neuromodulatory inputs would predominantly activate those of the metabotropic class. Slight rewording for clarity would be helpful here. p5; I137: is this supposed to refer to rates of discharge? section ending on I142: The discussion of the role of 5ht on the dorsal cord seems a little light - one might consider a slight expansion of this topic here so that the (predominantly) inhibitory effects of 5ht on sensory transmission can be appreciated. p7; I207-209: It may be nice to put this magnitude of change into perspective for the naive reader. If a healthy young adult has near complete activation of the biceps (near 100%; see pre-fatigue %VA in Fig 3) then we would not expect that they could gain more than a percent or two as they cannot have >complete activation (theoretical, I know). Other muscles, with lower %VA, may show greater change. p8; I235: put "i.e. the motoneurone's hysteresis" in parenthesis? p8; I236-237: Although seldomly considered in the literature, the threshold of the PIC can vary and may be a key contributor to MN recruitment. Please refer to the discussion of Afsharipour et al 2021 Section ending I375: I suggest updating this section with reference to the the emerging findings from both CJ Heckman's lab and from the second author's lab (references below). These findings, albeit preliminary in nature, both support the notion that afferent feedback may dampen the effects of PICs in humans. Pearcey et al. 2020 - Exploring the effects of Ia reciprocal inhibition on neuromodulatory commands in the human lower limb (<https://doi.org/10.1096/fasebj.2020.34.s1.09445>) Mesquita et al. 2022 - Effects of reciprocal inhibition and whole-body relaxation on persistent inward current estimated by two different methods (<https://doi.org/10.1113/JP282765>) Referee #3:

This review provides a comprehensive and balanced overview of the role of 5-HT on muscle activation. Insights from mammalian and from human studies are discussed. The authors do a commendable job of comparing the results from both animal and human studies and identify methodological challenges in testing hypotheses derived from animal work in human studies. The review will be very appreciated by the motor control community I only have minor comments: 1. I know that this review looks at voluntary muscle activation but I wonder if there are any insights about 5-HT perhaps modulating reflexive movements in humans and whether there could be a greater role in that type of movement based on the possibility that serotonergic systems are activated by novel or surprising stimuli 2. What about the role of 5-HT in mediating presynaptic inhibition of sensory afferents? Are there insights on this in humans, and does it relate to central fatigue? 3. If I can make a suggestion to add Bui et al. (2003) (work from Ken Rose lab) to the works cited in line 356. ----- REQUIRED ITEMS: - Please include an Abstract Figure. The Abstract Figure is a piece of artwork designed to give readers an immediate understanding of the Review Article and should summarise the main conclusions. If possible, the image should be easily 'readable' from left to right or top to bottom. It should show the physiological relevance of the Review so readers can assess the importance and content of the article. Abstract Figures should not merely recapitulate other figures in the Review. Please try to keep the diagram as simple as possible and without superfluous information that may distract from the main conclusion of the Review. Abstract Figures must be provided by authors no later than the revised manuscript stage and should be uploaded as a separate file during online submission labelled as File Type 'Abstract Figure'. Please ensure that you include the figure legend in the main article file. All Abstract Figures will be sent to a professional illustrator for redrawing and you may be asked to approve the redrawn figure before your paper is accepted. -Please upload separate high quality figure files via the submission form. -Author profile(s) must be uploaded via the submission form. Authors should submit a short biography (no more than 100 words for one author or 150 words in total for two authors) and a portrait photograph of the two leading authors on the paper. These should be uploaded, clearly labelled, with the manuscript submission. Any standard image format for the photograph is acceptable, but the resolution should be at least 300 dpi and preferably more. A group photograph of all authors is also acceptable, providing the biography for the whole group does not exceed 150 words. ----- END OF COMMENTS

Confidential Review

31-Mar-2022

The Journal of Physiology

<https://jp.msubmit.net>

JP-TR-2022-282565

Title: Voluntary activation of muscle in humans: does serotonergic neuromodulation matter?

Authors: Justin Kavanagh
Janet Taylor

Author Conflict: No competing interests declared

Author Contribution: Justin Kavanagh: Conception or design of the work; Acquisition or analysis or interpretation of data for the work; Drafting the work or revising it critically for important intellectual content; Final approval of the version to be published; Agreement to be accountable for all aspects of the work Janet Taylor: Conception or design of the work; Acquisition or analysis or interpretation of data for the work; Drafting the work or revising it critically for important intellectual content; Final approval of the version to be published; Agreement to be accountable for all aspects of the work

Running Title: Serotonin and voluntary muscle activation

Dual Publication: Figure 1B, 1C, and 1D: Our schematic of the serotonergic system and motoneurons is supported by cellular data

Disclaimer: This is a confidential document.

published in Cotel, Exley, Cragg, and Perrier JF, Proceedings of the National Academy of Sciences USA, 2013, 110(12):4774-9. This does not constitute dual publication as we use this representative image only to illustrate the main theme of the Topical review. Namely, that prior to our human work, cellular studies had revealed the dual role that serotonin plays in modulating motoneurone activity. The source publication is included in the figure caption. Figures 2 and 3: These are data sets that have been published by the lead author. They are also acknowledged in each figure caption. These figures should not be considered as dual publications as we are only using them to illustrate a series of published experiments which highlights the excitatory effect of serotonin during unfatigued muscle contractions (Figure 2), and the inhibitory effect of serotonin during fatigued muscle contractions (Figure 3). Figures are from: Kavanagh, McFarland and Taylor. Journal of Physiology, 2019, 597, 319-332. Thorstensen, Taylor, Tucker and Kavanagh. Journal of Physiology, 2020, 598, 2685-2701. Thorstensen, Taylor and Kavanagh. Journal of Neurophysiology, 2021, 125, 1279-1288. Henderson, Thorstensen, Morrison, Tucker and Kavanagh. Journal of Neurophysiology, 2022, 127, 27-37.

Funding: No funding: Justin J Kavanagh, N/A; No funding: Janet L Taylor, N/A

1 **JOURNAL OF PHYSIOLOGY TOPICAL REVIEW**

2

[revised manuscript text omitted]

5-HT receptor subtypes in the CNS refer to (Peroutka, 1990; Barnes & Sharp, 1999; Nichols
& Nichols, 2008; Perrier *et al.*, 2013).

From the brainstem, neurons from a group of serotonergic nuclei ascend to distribute across
many areas of the brain, including the frontal lobe where 5-HT can be released in the cerebral
cortex (Figure 1A). A high density of 5-HT fibres have been identified in motor areas of the
rat and primate cortex (Vertes, 1991; Wilson & Molliver, 1991), however the role of these
rostral raphe projections in motor activity has not been well-defined. Cortical structures
predominantly express receptors from the inhibitory 5-HT₁ family and to a lesser extent the
excitatory 5-HT₂ family (Pazos *et al.*, 1985; Pazos & Palacios, 1985; Celada *et al.*, 2013),
where 5-HT_{1A} is expressed extensively in laminae II/III of the motor cortex and 5-HT_{2A}
receptors are uniformly distributed across motor cortex laminae (Joyce *et al.*, 1993; Vitrac &
Benoit-Marand, 2017). It appears that depleting brain 5-HT in rats has the effect of reducing
the excitability of neural circuits in motor cortex (Scullion *et al.*, 2013), and administration of
selective serotonin reuptake inhibitors (SSRIs) enhances excitability in motor cortex circuits
in humans (Ilic *et al.*, 2002; Gerdelat-Mas *et al.*, 2005). Thus, a likely key role of 5-HT in the
motor cortex is modulation of facilitatory intracortical motor circuits. 5-HT also appears to
play a critical role in plasticity of motor circuits, as SSRI administration combined with either
non-invasive brain stimulation or motor practice can influence excitatory synaptic plasticity.
In particular, there is evidence to suggest enhanced long-term potentiation-like plasticity from
paired associative stimulation (Nitsche *et al.*, 2009; Batsikadze *et al.*, 2013), whereas changes
in plasticity from practicing motor tasks has been observed in most (Classen *et al.*, 1998;
Loubinoux *et al.*, 2002; Pleger *et al.*, 2004) but not all (McDonnell *et al.*, 2018) experiments
that have assessed cortical plasticity after SSRI ingestion. Functional magnetic resonance
imaging has identified several interconnected structures that are affected by enhanced 5-HT
concentration, where short-term drug-induced neuroplastic changes are associated with
enhanced activation of primary motor and premotor cortices, as well as posterior
supplementary motor areas (i.e. executive motor areas of the cortex) (Loubinoux *et al.*, 1999).

< insert Figure 1 >

Most of the work that has examined 5-HT effects on motor cortical activity has focused on
evoked responses and has not linked cortical changes to muscle performance. However,

considerably more is known about how 5-HT release on motor circuits in the spinal cord
directly affects motor activity. From the brainstem, the raphe-spinal pathway descends to the
spinal cord to form well-defined synapses with motoneurons, afferent neurons, and
interneurons (Figure 1A) (Ridet *et al.*, 1994; Alvarez *et al.*, 1998; Kawashima, 2018). There
are approximately 1500 serotonergic contacts on each motoneurone (Alvarez *et al.*, 1998),
and the intensity of 5-HT release from the raphe-spinal pathway, but not the ascending
pathway, is thought to correspond to the intensity of motor activity being performed by upper
limb (forelimb) and lower limb (hindlimb) muscles. This viewpoint has predominantly been
based on cat studies that show activity in the raphe nuclei is abolished during sleep (McGinty
& Harper, 1976; Cespuglio *et al.*, 1981), but near-linearly scaled to locomotion speed
(Veasey *et al.*, 1995; Jacobs & Fornal, 1997; Jacobs *et al.*, 2002). Incremental increases in
treadmill walking speed correspond to incremental increases in single raphe-spinal fibres
(Fornal *et al.*, 2006), which potentially points towards a contraction intensity dependent
mechanism of 5-HT release. On face value, this suggests a very useful motor control strategy,
in which strong contractions cause large release of 5-HT to amplify signals from synaptic
inputs. However, a phenomenon exists where the presence of too much 5-HT in the CNS can
limit the ability of motoneurons to activate muscle.

**Dual role of 5-HT on motoneurone activity**

There is widespread agreement that 5-HT promotes depolarisation in motoneurons, which
arises from several concomitant mechanisms. In particular, cellular and molecular
experiments have identified that 5-HT can facilitate a rectifying inward current (Takahashi &
Berger, 1990; Hsiao *et al.*, 1997), facilitate low threshold Ca^+ currents (Berger & Takahashi,
1990), or inhibit K^+ leak conductance (Elliott & Wallis, 1992; Perrier *et al.*, 2003) to
modulate neuronal excitability. 5-HT can also increase discharge rate via inhibition of slow
and medium afterhyperpolarisation (Hounsgaard *et al.*, 1988b; Hounsgaard & Kiehn, 1989;
Bayliss *et al.*, 1995; Wikstrom *et al.*, 1995; Grunnet *et al.*, 2004). Although 5-HT can inhibit
hyperpolarisation by decreasing the threshold for generating a Na^+ based action potentials
(Fedirchuk & Dai, 2004), 5-HT-mediated inhibition of hyperpolarisation also generates a Ca^+
dependent plateau potential by reducing a K^+ current responsible for slow
afterhyperpolarisation (Hounsgaard & Kiehn, 1989). In general, the functional role of 5-HT
on motoneurone activity has been revealed by slice preparations, partially intact animal
preparations, and simulations. With these approaches, it has been possible to examine the

interaction between physiological 5-HT concentrations and 5-HT receptor subtypes.
Electrical stimulation of the dorsolateral funiculus (DLF, analogous to the raphe-spinal
pathway) in integrated preparations of the adult turtle spinal cord has been shown to promote
the release of endogenous 5-HT into the ventral horn of the spinal cord (Delgado-Lezama *et*
*al.*, 1997; Perrier & Delgado-Lezama, 2005; Perrier & Cotel, 2008b). In this circumstance,
motoneurone recordings have confirmed that discharge rate increases with brief bursts of 5-
HT release (Figure 1B) (Hounsgaard & Kiehn, 1989; Perrier & Hounsgaard, 2003; Perrier &
Delgado-Lezama, 2005), and this increase is mediated by activation of 5-HT_{2B/C} receptors
located on the dendrites and the cell body of the neuron (Jackson & White, 1990; Cotel *et al.*,
2013).

In addition to 5-HT exerting excitatory effects on motoneurons, the presence of this
neuromodulator has also been observed to create an inhibitory effect on motor activity. Early
*in vivo* work that recorded motoneurone activity in cats observed that 5-HT injected in the
vicinity of the motoneurone increased hyperpolarisation, which caused a failure in the
motoneurone to fire (Phillis *et al.*, 1968). An underlying mechanism of 5-HT-mediated
inhibition of motoneurone activity was clarified in a series of elegant experiments by Perrier
and colleagues. Using preparations from the turtle spinal cord, it was revealed that the ability
of motoneurons to fire is inhibited when 5-HT is iontophoretically applied close to
motoneurone soma (Perrier & Hounsgaard, 2003) and when prolonged DLF stimulation
causes large endogenous release of 5-HT onto motoneurons (Figure 1C) (Cotel *et al.*, 2013;
Perrier *et al.*, 2017). The mechanism underlying 5-HT's dual excitatory and inhibitory effects
aligns with the compartmentalisation of 5-HT receptor subtypes expressed on motoneurons.
The monosynaptic connections that raphe-spinal neurons have with motoneurons causes
direct 5-HT release onto receptors in somatodendritic regions to enhance motoneurone
excitability. However, activation of 5-HT_{1A} receptors expressed in perisomatic regions
inhibits Na⁺ channels responsible for the genesis of action potentials, which has the effect of
preventing motoneurone firing (Perrier & Cotel, 2008a). Given that the inhibitory 5-HT_{1A}
receptors are only located on the axon initial segment, and the axon initial segment is devoid
of serotonergic innervation, activation of 5-HT_{1A} receptors can only occur if extracellular
concentrations of 5-HT are large enough to spill over onto the axon initial segment (Cotel *et*
*al.*, 2013; Perrier & Cotel, 2015; Perrier *et al.*, 2017). These findings provide a potential
cellular mechanism for central fatigue and provide a foundation to investigate if 5-HT
activity during prolonged bouts of physical activity reduces motor output in humans.

**Is serotonergic neuromodulation reflected in voluntary muscle activation in humans**

Although it is difficult to identify how individual neuromodulators contribute to motor
activity, simulations indicate that the net effect of maximal neuromodulation may be a three-
to five-fold amplification of the currents that the motoneurons receive from synaptic inputs
(Lee & Heckman, 2000; Hultborn *et al.*, 2003). Hence, the availability of 5-HT is critical to
enhancing the gain of motoneurone output, and ultimately, activating the muscle to produce
force. Data from healthy adults performing maximal effort contractions mostly support this
finding. However, the magnitude of response in human experiments is small (Figure 2A and
2B). Following the administration of the SSRI paroxetine to increase serotonin availability,
the force of maximal voluntary contractions (MVC) has been noted to increase 1.5-4.5% in
several experiments employing elbow flexion protocols. Similarly, voluntary activation
increased by 0.3% when measured using cortical stimulation (Thorstensen *et al.*, 2020) and
by ~1.3% when measured using motor nerve stimulation (Kavanagh *et al.*, 2019; Thorstensen
*et al.*, 2020). A finding of only small drug-related increases in voluntary activation is perhaps
not surprising, as increases in motoneurone excitation during near-maximal contraction
intensities produce only small changes in interpolated twitch amplitude (which is a key
variable in calculating voluntary activation) (Herbert & Gandevia, 1999). Nevertheless, an
enhanced ability to activate the muscle is consistent with 5-HT effects that are mediated by
excitatory somato-dendritic 5-HT₂ receptors on motoneurons. Consistent with this, the
administration of a 5-HT₂ competitive antagonist leads to the opposite effect on maximal
force generation. Maximal elbow flexion force declined by ~3% after the administration of
cyproheptadine (Thorstensen *et al.*, 2021). Antagonism of the 5-HT₂ receptor also leads to a
reduction in rate of force development when performing rapid dorsiflexions, which is
mediated by reduced firing rates in motor units of the tibialis anterior (Goodlich *et al.*, 2022).
Thus, enhancing 5-HT availability during strong unfatigued contractions increases force
generation, whereas blocking 5-HT₂ receptor activity decreases force generation via neural
mechanisms in healthy individuals.

< insert Figure 2 >

Persistent inward currents (PIC) play a critical role in setting the gain of motoneurons. PICs
are mediated by voltage-gated sodium channels ($\text{Na}_v1.1$ and $\text{Na}_v1.6$) and L-type voltage-
gated calcium channels ($\text{Ca}_v1.2$ and $\text{Ca}_v1.3$) on somato-dendritic surfaces of motoneurons
(Schwindt & Crill, 1977; Schwindt & Crill, 1980; Heckmann *et al.*, 2005). In mammals, there
is believed to be equal contributions from slow activating L-type Ca^{2+} current and a fast
activating persistent Na^+ current to the PIC (Heckman *et al.*, 2008a). Moreover, PICs are
strongly influenced by monoamines. Activation of PICs causes a remarkable amplification of
depolarising drive to the motoneuron and evokes a strong acceleration of motoneuron firing
rate (Hounsgaard *et al.*, 1988a; Bennett *et al.*, 1998; Lee & Heckman, 1998). Thus, it might
be expected that PICs are responsible for 5-HT mediated increases in voluntary activation
during MVCs. In humans, the amplitude of PIC activation can only be estimated. The
accepted technique for this estimation for an individual motoneuron compares the estimated
synaptic input that recruits a motoneuron to that at its derecruitment i.e. the motoneuron's
hysteresis. PICs provide little drive to the motoneuron at recruitment as initiation of an
action potential and the activation of PICs occur at a similar membrane potential. By
comparison, PICs are fully active after the motoneuron has been firing for ~ 0.5 - 1.0 s (Lee
& Heckman, 1996; Heckman *et al.*, 2008b). Thus, at derecruitment, PICs contribute part of
the depolarising current for motoneuron firing so that less synaptic input is required.

The paired motor unit technique uses the firing rate of low threshold motor units, that already
have PICs fully active, as estimates of synaptic input to other motor units in the same pool.
The measure that is calculated is known as ΔF (difference in frequency of firing of the lower
threshold motor unit at recruitment and derecruitment of the higher threshold motor unit).
Discharge rates are recorded from intramuscular electromyography electrodes (Gorassini *et al.*,
2002; Udina *et al.*, 2010; Foley & Kalmar, 2019) or extracted from high-density
electromyography arrays placed over the muscle (Afsharipour *et al.*, 2020; Hassan *et al.*,
2020). The idea that the influence of serotonin on human motor performance is mediated via
PICs is supported by pharmacological studies in a small number of healthy participants ($n =$
3), in whom ΔF increased with ingestion of an SSRI or decreased with a 5-HT₂ receptor
antagonist (D'Amico *et al.*, 2013). Nonetheless, because ΔF has only been calculated during
submaximal ramped-shaped contractions, it is unknown if PICs are directly linked to
enhanced MVCs in humans. However, it is known that PICs associated with soleus and
gastrocnemius medialis increase from 10% to 20% MVC with additional increases to 30%
MVC for soleus, which suggests that PIC activation is linked to voluntary drive and hence 5-

HT release (Orssatto *et al.*, 2021b). Reductions in motoneurone PIC may also provide a
mechanism for loss of force production from skeletal muscle. In younger populations, there is
evidence that ΔF reduces up to 25% for soleus motoneurons following passive stretching of
the plantarflexors, which may explain some of the reductions in force associated with
stretching protocols (Trajano *et al.*, 2020). In older populations, ΔF is lower than in young
adults for soleus and tibialis anterior (Orssatto *et al.*, 2021a), and biceps and triceps brachii
(Hassan *et al.*, 2021), and may contribute to age-related reductions in motoneurone firing
rates and muscle force. However, the mechanisms underlying the reductions in PIC activity
are not known. Conceivably, monoamine release onto the motoneurons could be reduced or
the motoneuronal response to monoamines could be altered by changes in receptors.
Alternatively, PICs are sensitive to inhibitory synaptic input and could be turned off by added
inhibition.

**Is prolonged release of serotonin a mechanism of fatigue in humans?**

[revised manuscript text omitted]

**Competing interests and funding**

The authors declare that no competing interests exist for this work and no funding was
received to perform this work.

**Author contributions**

Both authors contributed to the conception and the design of this work, as well as the drafting
and final approval of the manuscript.

**References**

- Afsharipour B, Manzur N, Duchcherer J, Fenrich KF, Thompson CK, Negro F, Quinlan KA,
Bennett DJ & Gorassini MA. (2020). Estimation of self-sustained activity produced
by persistent inward currents using firing rate profiles of multiple motor units in
humans. *Journal of Neurophysiology* **124**, 63-85.
Alexander S, Mathie A & Peters J. (2011). Ligand-gated ion channels. *British Journal of*
*Pharmacology* **164**, S115-S135.
Alvarez FJ, Pearson JC, Harrington D, Dewey D, Torbeck L & Fyffe RE. (1998).
Distribution of 5-hydroxytryptamine-immunoreactive boutons on alpha-motoneurons
in the lumbar spinal cord of adult cats. *J Comp Neurol* **393**, 69-83.
Barnes NM & Sharp T. (1999). A review of central 5-HT receptors and their function.
*Neuropharmacology* **38**, 1083-1152.
Batsikadze G, Paulus W, Kuo MF & Nitsche MA. (2013). Effect of serotonin on paired
associative stimulation-induced plasticity in the human motor cortex.
*Neuropsychopharmacology* **38**, 2260-2267.
Bayliss DA, Umemiya M & Berger AJ. (1995). Inhibition of N- and P-type calcium currents
and the after-hyperpolarization in rat motoneurons by serotonin. *J Physiol* **485 (Pt**
**3)**, 635-647.
Bennett DJ, Hultborn H, Fedirchuk B & Gorassini M. (1998). Synaptic Activation of Plateaus
in hindlimb motoneurons of decerebrate Cats. *Journal of Neurophysiology* **80**, 2023-
2037.
Berger AJ & Takahashi T. (1990). Serotonin enhances a low-voltage-activated calcium
current in rat spinal motoneurons. *J Neurosci* **10**, 1922-1928.
Celada P, Puig MV & Artigas F. (2013). Serotonin modulation of cortical neurons and
networks. *Front Integr Neurosci* **7**, 25.
Cespuglio R, Faradji H, Gomez ME & Jouvet M. (1981). Single unit recordings in the nuclei
raphe dorsalis and magnus during the sleep-waking cycle of semi-chronic prepared
cats. *Neurosci Lett* **24**, 133-138.
Classen J, Liepert J, Wise SP, Hallett M & Cohen LG. (1998). Rapid plasticity of human
cortical movement representation induced by practice. *J Neurophysiol* **79**, 1117-1123.

- Cotel F, Exley R, Cragg SJ & Perrier JF. (2013). Serotonin spillover onto the axon initial
segment of motoneurons induces central fatigue by inhibiting action potential
initiation. *Proc Natl Acad Sci U S A* **110**, 4774-4779.
D'Amico JM, Butler AA, Heroux ME, Cotel F, Perrier JM, Butler JE, Gandevia SC & Taylor
JL. (2017). Human motoneurone excitability is depressed by activation of serotonin
1A receptors with buspirone. *J Physiol* **595**, 1763-1773.
D'Amico JM, Condliffe EG, Martins KJ, Bennett DJ & Gorassini MA. (2014). Recovery of
neuronal and network excitability after spinal cord injury and implications for
spasticity. *Front Integr Neurosci* **8**, 36.
D'Amico JM, Murray KC, Li Y, Chan KM, Finlay MG, Bennett DJ & Gorassini MA. (2013).
Constitutively active 5-HT₂/α receptors facilitate muscle spasms after human
spinal cord injury. *J Neurophysiol* **109**, 1473-1484.
Delgado-Lezama R, Perrier JF, Nedergaard S, Svirskis G & Hounsgaard J. (1997).
Metabotropic synaptic regulation of intrinsic response properties of turtle spinal
motoneurons. *J Physiol* **504 (Pt 1)**, 97-102.
Eccles JC & McGeer PL. (1979). Ionotropic and metabotropic neurotransmission *Trends in*
*Neuroscience* **23**, 39-40.
Elliott P & Wallis DI. (1992). Serotonin and L-norepinephrine as mediators of altered
excitability in neonatal rat motoneurons studied in vitro. *Neuroscience* **47**, 533-544.
Fedirchuk B & Dai Y. (2004). Monoamines increase the excitability of spinal neurones in the
neonatal rat by hyperpolarizing the threshold for action potential production. *J Physiol*
**557**, 355-361.
Foley RCA & Kalmar JM. (2019). Estimates of persistent inward current in human motor
neurons during postural sway. *Journal of Neurophysiology* **122**, 2095-2110.
Fornal CA, Martin-Cora FJ & Jacobs BL. (2006). "Fatigue" of medullary but not
mesencephalic raphe serotonergic neurons during locomotion in cats. *Brain Res* **1072**,
55-61.
Gerdelat-Mas A, Loubinoux I, Tombari D, Rascol O, Chollet F & Simonetta-Moreau M.
(2005). Chronic administration of selective serotonin reuptake inhibitor (SSRI)
paroxetine modulates human motor cortex excitability in healthy subjects.
*Neuroimage* **27**, 314-322.

- Goodlich BI, Horan SA & Kavanagh JJ. (2022). Blockade of 5-HT₂ receptors suppresses rate
of torque development and motor unit discharge rate during rapid contractions. *J*
*Neurophysiol* **127**, 150-160.
Gorassini M, Yang JF, Siu M & Bennett DJ. (2002). Intrinsic activation of human
motoneurons: Possible contribution to motor unit excitation. *Journal of*
*Neurophysiology* **87**, 1850-1858.
Grunnet M, Jespersen T & Perrier JF. (2004). 5-HT_{1A} receptors modulate small-conductance
Ca²⁺-activated K⁺ channels. *J Neurosci Res* **78**, 845-854.
Hassan A, Thompson CK, Negro F, Cummings M, Powers RK, Heckman CJ, Dewald JPA &
McPherson LM. (2020). Impact of parameter selection on estimates of motoneuron
excitability using paired motor unit analysis. *J Neural Eng* **17**.
Hassan AS, Fajardo ME, Cummings M, McPherson LM, Negro F, Dewald JPA, Heckman CJ
& Pearcey GE. (2021). Estimates of persistent inward currents are reduced in upper
limb motor units of older adults. *J Physiol*.
Heckman CJ & Binder MD. (1991). Analysis of Ia-inhibitory synaptic input to cat spinal
motoneurons evoked by vibration of antagonist muscles. *J Neurophysiol* **66**, 1888-
1893.
Heckman CJ, Hyngstrom AS & Johnson MD. (2008a). Active properties of motoneurone
dendrites: diffuse descending neuromodulation, focused local inhibition. *J Physiol*
**586**, 1225-1231.
Heckman CJ, Johnson M, Mottram C & Schuster J. (2008b). Persistent inward currents in
spinal motoneurons and their influence on human motoneuron firing patterns.
*Neuroscientist* **14**, 264-275.
Heckmann CJ, Gorassini MA & Bennett DJ. (2005). Persistent inward currents in
motoneuron dendrites: implications for motor output. *Muscle Nerve* **31**, 135-156.
Henderson TT, Thorstensen JR, Morrison S, Tucker MG & Kavanagh JJ. (2022).
Physiological tremor is suppressed and force steadiness is enhanced with increased
availability of serotonin regardless of muscle fatigue. *J Neurophysiol* **127**, 27-37.
Herbert RD & Gandevia SC. (1999). Twitch interpolation in human muscles: mechanisms
and implications for measurement of voluntary activation. *J Neurophysiol* **82**, 2271-
2283.
Hery F, Faudon M & Ternaux JP. (1982). In vivo release of serotonin in two raphe nuclei
(raphe dorsalis and magnus) of the cat. *Brain Res Bull* **8**, 123-129.

Hornung JP. (2003). The human raphe nuclei and the serotonergic system. *J Chem Neuroanat*
**26**, 331-343.
Hounsgaard J, Hultborn H, Jespersen B & Kiehn O. (1988a). Bistability of Alpha-
Motoneurons in the Decerebrate Cat and in the Acute Spinal Cat after Intravenous 5-
Hydroxytryptophan. *J Physiol-London* **405**, 345-367.
Hounsgaard J & Kiehn O. (1989). Serotonin-induced bistability of turtle motoneurons
caused by a nifedipine-sensitive calcium plateau potential. *J Physiol* **414**, 265-282.
Hounsgaard J, Kiehn O & Mintz I. (1988b). Response properties of motoneurons in a slice
preparation of the turtle spinal cord. *J Physiol* **398**, 575-589.
Hoyer D & Martin G. (1997). 5-HT receptor classification and nomenclature: towards a
harmonization with the human genome. *Neuropharmacology* **36**, 419-428.
Hsiao CF, Trueblood PR, Levine MS & Chandler SH. (1997). Multiple effects of serotonin
on membrane properties of trigeminal motoneurons in vitro. *J Neurophysiol* **77**, 2910-
2924.
Hultborn H, Denton ME, Wienecke J & Nielsen JB. (2003). Variable amplification of
synaptic input to cat spinal motoneurons by dendritic persistent inward current. *J*
*Physiol* **552**, 945-952.
Hyngstrom A, Johnson M, Schuster J & Heckman CJ. (2008). Movement-related receptive
fields of spinal motoneurons with active dendrites. *J Physiol* **586**, 1581-1593.
Hyngstrom AS, Johnson MD, Miller JF & Heckman CJ. (2007). Intrinsic electrical properties
of spinal motoneurons vary with joint angle. *Nat Neurosci* **10**, 363-369.
Ilic TV, Korchounov A & Ziemann U. (2002). Complex modulation of human motor cortex
excitability by the specific serotonin re-uptake inhibitor sertraline. *Neurosci Lett* **319**,
116-120.
Imai H, Steindler DA & Kitai ST. (1986). The organization of divergent axonal projections
from the midbrain raphe nuclei in the rat. *J Comp Neurol* **243**, 363-380.
Jackson DA & White SR. (1990). Receptor subtypes mediating facilitation by serotonin of
excitability of spinal motoneurons. *Neuropharmacology* **29**, 787-797.
Jacobs BL & Fornal CA. (1997). Serotonin and motor activity. *Curr Opin Neurobiol* **7**, 820-
825.

Jacobs BL, Martin-Cora FJ & Fornal CA. (2002). Activity of medullary serotonergic neurons
in freely moving animals. *Brain Res Brain Res Rev* **40**, 45-52.
Joyce JN, Shane A, Lexow N, Winokur A, Casanova MF & Kleinman JE. (1993). Serotonin
uptake sites and serotonin receptors are altered in the limbic system of schizophrenics.
*Neuropsychopharmacology* **8**, 315-336.
Kavanagh JJ, McFarland AJ & Taylor JL. (2019). Enhanced availability of serotonin
increases activation of unfatigued muscle but exacerbates central fatigue during
prolonged sustained contractions. *J Physiol* **597**, 319-332.
Kawashima T. (2018). The role of the serotonergic system in motor control. *Neurosci Res*
**129**, 32-39.
Kernell D. (2006). *The motoneurone and its muscle fibres*. Oxford University Press, Oxford ;
New York.
Klass M, Duchateau J, Rabec S, Meeusen R & Roelands B. (2016). Noradrenaline Reuptake
Inhibition Impairs Cortical Output and Limits Endurance Time. *Med Sci Sports Exerc*
**48**, 1014-1023.
Klass M, Roelands B, Levenez M, Fontenelle V, Pattyn N, Meeusen R & Duchateau J.
(2012). Effects of noradrenaline and dopamine on supraspinal fatigue in well-trained
men. *Med Sci Sports Exerc* **44**, 2299-2308.
Kuo JJ, Lee RH, Johnson MD, Heckman HM & Heckman CJ. (2003). Active dendritic
integration of inhibitory synaptic inputs in vivo. *J Neurophysiol* **90**, 3617-3624.
Lee RH & Heckman CJ. (1996). Influence of voltage-sensitive dendritic conductances on
bistable firing and effective synaptic current in cat spinal motoneurons in vivo. *J*
*Neurophysiol* **76**, 2107-2110.
Lee RH & Heckman CJ. (1998). Bistability in spinal motoneurons in vivo: Systematic
variations in persistent inward currents. *Journal of Neurophysiology* **80**, 583-593.
Lee RH & Heckman CJ. (2000). Adjustable amplification of synaptic input in the dendrites of
spinal motoneurons in vivo. *J Neurosci* **20**, 6734-6740.
Loubinoux I, Boulanouar K, Ranjeva JP, Carel C, Berry I, Rascol O, Celsis P & Chollet F.
(1999). Cerebral functional magnetic resonance imaging activation modulated by a
single dose of the monoamine neurotransmission enhancers fluoxetine and fenozolone
during hand sensorimotor tasks. *J Cereb Blood Flow Metab* **19**, 1365-1375.

Loubinoux I, Pariente J, Rascol O, Celsis P & Chollet F. (2002). Selective serotonin reuptake
inhibitor paroxetine modulates motor behavior through practice. A double-blind,
placebo-controlled, multi-dose study in healthy subjects. *Neuropsychologia* **40**, 1815-
1821.
Lucas JJ & Hen R. (1995). New players in the 5-HT receptor field: genes and knockouts.
*Trends Pharmacol Sci* **16**, 246-252.
Marvin G, Sharma A, Aston W, Field C, Kendall MJ & Jones DA. (1997). The effects of
buspirone on perceived exertion and time to fatigue in man. *Exp Physiol* **82**, 1057-
1060.
McDonnell MN, Zipser C, Darmani G, Ziemann U & Muller-Dahlhaus F. (2018). The effects
of a single dose of fluoxetine on practice-dependent plasticity. *Clin Neurophysiol* **129**,
1349-1356.
McGinty DJ & Harper RM. (1976). Dorsal raphe neurons: depression of firing during sleep in
cats. *Brain Res* **101**, 569-575.
Meeusen R, Piacentini MF, Van Den Eynde S, Magnus L & De Meirleir K. (2001). Exercise
performance is not influenced by a 5-HT reuptake inhibitor. *International Journal of*
*Sports Medicine* **22**, 329-336.
Murray KC, Stephens MJ, Ballou EW, Heckman CJ & Bennett DJ. (2011). Motoneuron
excitability and muscle spasms are regulated by 5-HT_{2B} and 5-HT_{2C} receptor
activity. *J Neurophysiol* **105**, 731-748.
Newsholme EA, Acworth IN & Blomstrand E. (1987). Amino acids, brain neurotransmitters
and a functional link between muscle and brain that is important in sustained exercise.
In *Advances in myochemistry*, ed. Benzi G, pp. 127-133. John Libbey, London.
Newsholme EA & Blomstrand E. (2006). Branched-chain amino acids and central fatigue. *J*
*Nutr* **136**, 274S-276S.
Nichols DE & Nichols CD. (2008). Serotonin receptors. *Chem Rev* **108**, 1614-1641.
Nitsche MA, Kuo MF, Karrasch R, Wachter B, Liebetanz D & Paulus W. (2009). Serotonin
affects transcranial direct current-induced neuroplasticity in humans. *Biol Psychiatry*
**66**, 503-508.
Orssatto LBR, Borg DN, Blazeovich AJ, Sakugawa RL, Shield AJ & Trajano GS. (2021a).
Intrinsic motoneuron excitability is reduced in soleus and tibialis anterior of older
adults. *Geroscience* **43**, 2719-2735.

Orssatto LBR, Mackay K, Shield AJ, Sakugawa RL, Blazevich AJ & Trajano GS. (2021b).
Estimates of persistent inward currents increase with the level of voluntary drive in
low-threshold motor units of plantar flexor muscles. *Journal of Neurophysiology* **125**,
1746-1754.
Pazos A, Cortes R & Palacios JM. (1985). Quantitative autoradiographic mapping of
serotonin receptors in the rat brain. II. Serotonin-2 receptors. *Brain Res* **346**, 231-249.
Pazos A & Palacios JM. (1985). Quantitative autoradiographic mapping of serotonin
receptors in the rat brain. I. Serotonin-1 receptors. *Brain Res* **346**, 205-230.
Peroutka SJ. (1990). 5-Hydroxytryptamine Receptor Subtypes. *Pharmacology & Toxicology*
**67**, 373-383.
Perrier JF, Alaburda A & Hounsgaard J. (2003). 5-HT_{1A} receptors increase excitability of
spinal motoneurons by inhibiting a TASK-1-like K⁺ current in the adult turtle. *J*
*Physiol* **548**, 485-492.
Perrier JF & Cotel F. (2008a). Serotonin differentially modulates the intrinsic properties of
spinal motoneurons from the adult turtle. *J Physiol* **586**, 1233-1238.
Perrier JF & Cotel F. (2008b). Serotonin differentially modulates the intrinsic properties of
spinal motoneurons from the adult turtle. *J Physiol* **586**, 1233-1238.
Perrier JF & Cotel F. (2015). Serotonergic modulation of spinal motor control. *Curr Opin*
*Neurobiol* **33**, 1-7.
Perrier JF & Delgado-Lezama R. (2005). Synaptic release of serotonin induced by
stimulation of the raphe nucleus promotes plateau potentials in spinal motoneurons of
the adult turtle. *J Neurosci* **25**, 7993-7999.
Perrier JF & Hounsgaard J. (2003). 5-HT₂ receptors promote plateau potentials in turtle
spinal motoneurons by facilitating an L-type calcium current. *J Neurophysiol* **89**, 954-
959.
Perrier JF, Rasmussen HB, Christensen RK & Petersen AV. (2013). Modulation of the
intrinsic properties of motoneurons by serotonin. *Curr Pharm Des* **19**, 4371-4384.
Perrier JF, Rasmussen HB, Jorgensen LK & Berg RW. (2017). Intense activity of the raphe
spinal pathway depresses motor activity via a serotonin dependent mhohechanism.
*Front Neural Circuits* **11**, 111.

- Phillis JW, Tebecis AK & York DH. (1968). Depression of spinal motoneurons by
noradrenaline, 5-hydroxytryptamine and histamine. *Eur J Pharmacol* **4**, 471-475.
Pleger B, Schwenkreis P, Grunberg C, Malin JP & Tegenthoff M. (2004). Fluoxetine
facilitates use-dependent excitability of human primary motor cortex. *Clin*
*Neurophysiol* **115**, 2157-2163.
Pollak Dorocic I, Furth D, Xuan Y, Johansson Y, Pozzi L, Silberberg G, Carlen M & Meletis
723 K. (2014). A whole-brain atlas of inputs to serotonergic neurons of the dorsal and
724 median raphe nuclei. *Neuron* **83**, 663-678.
Powers RK, Turker KS & Binder MD. (2002). What can be learned about motoneurone
properties from studying firing patterns? *Adv Exp Med Biol* **508**, 199-205.
Ridet JL, Tamir H & Privat A. (1994). Direct immunocytochemical localization of 5-
hydroxytryptamine receptors in the adult rat spinal cord: a light and electron
microscopic study using an anti-idiotypic antiserum. *J Neurosci Res* **38**, 109-121.
Roelands B, Goekint M, Buyse L, Pauwels F, De Schutter G, Piacentini F, Hasegawa H,
Watson P & Meeusen R. (2009). Time trial performance in normal and high ambient
temperature: is there a role for 5-HT? *European Journal of Applied Physiology* **107**,
119-126.
Schwindt P & Crill WE. (1977). A persistent negative resistance in cat lumbar motoneurons.
*Brain Res* **120**, 173-178.
Schwindt PC & Crill WE. (1980). Properties of a persistent inward current in normal and
TEA-injected motoneurons. *J Neurophysiol* **43**, 1700-1724.
Scullion K, Boychuk JA, Yamakawa GR, Rodych JTG, Nakanishi ST, Seto A, Smith VM,
McCarthy RW, Whelan PJ, Antle MC, Pittman QJ & Teskey GC. (2013). Serotonin
1A receptors alter expression of movement representations. *J Neurosci* **33**, 4988-
4999.
Sogaard K, Gandevia SC, Todd G, Petersen NT & Taylor JL. (2006). The effect of sustained
low-intensity contractions on supraspinal fatigue in human elbow flexor muscles. *J*
*Physiol* **573**, 511-523.
Strachan AT, Leiper JB & Maughan RJ. (2004). Paroxetine administration failed to influence
human exercise capacity, perceived effort or hormone responses during prolonged
exercise in a warm environment. *Experimental Physiology* **89**, 657-664.

- Strüder HK, Hollmann W, Platen P, Donike M, Gotzmann A & Weber K. (1998). Influence
of paroxetine, branched-chain amino acids and tyrosine on neuroendocrine system
responses and fatigue in humans. *Hormone and Metabolic Research* **30**, 188-194.
Takahashi T & Berger AJ. (1990). Direct excitation of rat spinal motoneurons by serotonin.
*J Physiol* **423**, 63-76.
Teixeira-Coelho F, Uendele-Pinto JP, Serafim AC, Wanner SP, de Matos Coelho M &
Soares DD. (2014). The paroxetine effect on exercise performance depends on the
aerobic capacity of exercising individuals. *Journal of Sports Science and Medicine*
**13**, 232-243.
Terry N & Margolis KG. (2017). Serotonergic Mechanisms Regulating the GI Tract:
Experimental Evidence and Therapeutic Relevance. *Handb Exp Pharmacol* **239**, 319-
342.
Thompson AJ & Lummis SC. (2006). 5-HT₃ receptors. *Curr Pharm Des* **12**, 3615-3630.
Thorstensen JR, Taylor JL & Kavanagh JJ. (2021). Human corticospinal-motoneuronal
output is reduced with 5-HT₂ receptor antagonism. *J Neurophysiol* **125**, 1279-1288.
Thorstensen JR, Taylor JL, Tucker MG & Kavanagh JJ. (2020). Enhanced serotonin
availability amplifies fatigue perception and modulates the TMS-induced silent period
during sustained low-intensity elbow flexions. *J Physiol* **598**, 2685-2701.
Thorstensen JR, Tucker MG & Kavanagh JJ. (2018). Antagonism of the D₂ dopamine
receptor enhances tremor but reduces voluntary muscle activation in humans.
*Neuropharmacology* **141**, 343-352.
Trajano GS, Taylor JL, Orssatto LBR, McNulty CR & Blazevich AJ. (2020). Passive muscle
stretching reduces estimates of persistent inward current strength in soleus motor
units. *J Exp Biol* **223**.
Udina E, D'Amico J, Bergquist AJ & Gorassini MA. (2010). Amphetamine increases
persistent inward currents in human motoneurons estimated from paired motor-unit
activity. *J Neurophysiol* **103**, 1295-1303.
Veasey SC, Fornal CA, Metzler CW & Jacobs BL. (1995). Response of serotonergic caudal
raphe neurons in relation to specific motor activities in freely moving cats. *J Neurosci*
**15**, 5346-5359.
Vertes RP. (1991). A PHA-L analysis of ascending projections of the dorsal raphe nucleus in
the rat. *J Comp Neurol* **313**, 643-668.

[revised manuscript text omitted]

REVIEWING EDITOR

This manuscript has been reviewed by 3 expert reviewers. They unanimously enjoyed reading this review, noted its scholarship qualities and ultimately recommend publication. Several minor suggestions and clarifications are proposed and they appear to be expert and reasonable. As such, the authors should have no difficulties in addressing them in a timely manner.

We thank the Reviewing Editor for the positive comments regarding our manuscript. We have addressed each referee in turn and have highlighted in our manuscript where amendments have been made.

REFeree #1

Kavanagh and Taylor provide a timely review with the goal of highlighting and summarizing recent work in humans that describe roles and underlying mechanisms for 5HT and target receptors in the voluntary activation of muscle in humans. To accomplish this end, they place their human work in the context of multiple animal models that describe how serotonin modulates intrinsic properties and ion channels of motoneurons. The review draws the conclusions that studying 5HT control of muscle activation in humans might not be as straight forward as initially thought and propose/discuss several potential challenges. Because of the cross-human-animal comparison, this review will be of interest and useful to those studying neural control of muscle in humans and likewise, will be of interest to those studying mechanisms governing motoneuron excitability in animal models. This review is well written and was enjoyable to read. That being said, I have several comments that I think should be addressed. It is my hope that these comments will strengthen the review.

We thank Referee #1 for these positive comments. The referee has provided exceptional comments for us to consider, and we have incorporated nearly all suggestions in our revised manuscript. We would, however, like to preface our responses by pointing out that the majority of recommendations were for the animal and cellular sections of the manuscript. We had to carefully consider how much additional information to include in these sections so our Topical Review retains its focus on human muscle activation.

Of note, the manuscript was not submitted with line numbers but was generated by the editor upon request. To ensure line number references in this review match that of the document, Page 1 Line 1 starts with 'Journal of Physiology Topical Review', Page 2 Line 31: 'Abstract', Page 2 Line 47: 'Introduction'...etc.

We have included line numbers in our revised manuscript.

Page 2 Line 38-40: the statement saying 5HT can have profound effects on force generation during muscle contractions is a bit of an overstatement as the review highlights recent work that shows a very modest change MVC following manipulation of the serotonergic system. While I can agree that 5HT has a potent effect on firing rates, this doesn't seem to translate to huge changes in maximal muscle force - which is

interesting and discussed in the review. I think this discrepancy should either be highlighted in the abstract to provide context for the title of the review. Alternatively, I suggest toning back this statement.

To keep the abstract succinct, we have decided to tone back this statement. We feel that drawing attention to this particular discrepancy may detract from the many other comparisons that are made throughout the review. Our statement now reads, “Serotonin (5-HT) is a potent regulator of neuronal firing rates which can influence the force that can be generated by muscles during voluntary contractions.

Line 73: Why has serotonergic modulation been so extensively studied? While I can agree that it is a potent modulatory system in the control of spinal circuits, it is far from the only one. I think that this would be worth highlighting and would be useful to provide reviews that describe roles for other neuromodulators in the control of spinal circuits/motoneuron excitability (eg. Cholinergic, dopaminergic, noradrenergic, etc.).

We agree that several neuromodulator systems are involved in the regulation of motor circuits in the CNS. We have included the following information in the revised introduction with a view of remaining focussed on the serotonergic system, but also acknowledging that other neurotransmitter systems in the CNS contribute to muscle activation:

“There are several neuromodulator systems that regulate the excitability of cortical and spinal motor circuits, and many excellent reviews have been written that describe the effects of each system on motor activity (e.g. the cholinergic system (Jones, 2008; Deffains & Bergman, 2015; Naicker et al., 2017; Mille et al., 2021), the dopaminergic system (Sharples et al., 2014; Ikeda et al., 2015; Klaus et al., 2019; Arber & Costa, 2022), the noradrenergic system (Fung et al., 1994; Balaban, 2002; Benarroch, 2018)). Our topical review will provide a unique summary regarding how serotonergic neuromodulation contributes to voluntary activation of muscle in humans.”

Line 88: While the effects of ionotropic neurotransmitters may be terminated by removal of the ligand, the effects of neuromodulators, such as 5HT on neuronal function may not necessarily be terminated by reuptake of 5HT given that activation of GPCRs can elicit long lasting changes in ion channel function.

We agree that this is an important consideration that should be included in the manuscript. We have now modified our text to state:

“It is often suggested that the termination of 5-HT effects at the synapse occurs with reuptake into pre-synaptic terminals or glial cells via monoamine transporters. However, serotonergic effects on neuronal function may not necessarily end with reuptake of 5-HT, as activation of G protein-coupled receptors can elicit long lasting changes in ion channel function (Pavlos & Friedman, 2017)”.

Line 125: it is not clear what is meant by evoked responses. Would this be using TMS in humans or in animal models using other approaches?

This was a poorly worded sentence, and we have changed the text to read, “Most of the work that has examined 5-HT effects on motor cortical activity has employed magnetic or electrical stimulation techniques to explore cortical plasticity, or neuroimaging techniques to map changes in regional activity due to changes in neurotransmitter concentration.”

Line 137: worth mentioning is work from Brian Noga's Lab that used voltammetry to show increases in 5HT that occurs on the time scale of seconds during MLR-evoked fictive locomotion in cats (<https://doi.org/10.3389/fncir.2017.00059>). Interestingly, 5HT levels begin to increase prior to the onset of fictive locomotion which is supportive of a role for 5HT in setting the network tone for locomotion.

We thank Reviewer 1 for highlighting this work, as it has provided a valuable piece of information for our manuscript. Although the reviewer has suggested that this study could be included in our discussion of contraction intensity, we believe that it is best suited several sentences earlier where the onset of 5-HT release complements our anatomical descriptions of the 5-HT system. Our revised sentence now reads, “There are approximately 1500 serotonergic contacts on each motoneurone (Alvarez et al., 1998), where the release of 5-HT occurs 0.5-1.0 s following the onset of detectable neural activation of locomotor muscles (Noga et al., 2017)”.

Line 146-150: the three mechanisms described would collectively increase neuronal excitability. I think it is worthwhile using precise language to highlight this here. Eg change modulate on Line 150 to increase.

We have changed the term ‘modulate’ to ‘increase’ in the revised manuscript.

Line 152: It is unclear what is exactly meant by 'inhibition of hyperpolarization'. It is not clear how reducing spike threshold would inhibit hyperpolarization. Lines 152-156 should be rethought and reworded for clarity.

We have modified the text in the former line 152 to read, “5-HT can also increase discharge rate via mechanisms that reduce the amplitude of slow and medium afterhyperpolarisation phases that follow the action potential (Hounsgaard et al., 1988; Hounsgaard & Kiehn, 1989; Bayliss et al., 1995; Wikstrom et al., 1995; Grunnet et al., 2004)”.

Similarly, we have reworded the sentences that follow this statement to improve clarity in our discussion, “Although 5-HT can reduce hyperpolarisation by decreasing the threshold for generating a Na⁺ based action potentials (Fedirchuk & Dai, 2004), 5-HT also generates a Ca⁺ dependent plateau potential by reducing a K⁺ current responsible for slow afterhyperpolarisation (Hounsgaard & Kiehn, 1989).

Line 160: the DLF is not a specific analogue of the raphe-spinal pathway, but does contain descending serotonergic fibres, in addition to many other. This might be misleading as written.

We have changed our description of the dorsolateral funiculus to, “DLF, contains descending serotonergic fibres”.

Line 167: which neuron?

We have changed the term ‘neuron’ to ‘motoneurone’ in the revised manuscript.

Line 169: An additional inhibitory and activity dependent mechanism mediated by dynamic sodium potassium ATPase pumps has also been demonstrated by the Sillar lab for control of motoneuron activity in Xenopus tadpoles (Zhang et al. 2012; 2015). Importantly, this inhibitory pump is under the control of 5HT receptors (Hachoumi et al., 2022). Similar inhibitory pump dynamics have also been demonstrated in mammalian motoneurons and can be modulated by dopamine (Picton et al., 2016). These studies are worth highlighting for intrinsic cellular mechanisms that could contribute to the inhibitory (and activity dependent) regulation of motoneuron excitability.

We have included the following sentences to the end of this paragraph:

“Finally, other activity-dependent mechanisms may be influenced by serotonin. Sodium-potassium pump activity can cause inhibitory effects on motoneurone discharge during rhythmic activity (Zhang & Sillar, 2012; Zhang et al., 2015; Picton and Sillar, 2016). In tadpoles, this effect can be modulated bidirectionally through 5-HT_{2A} and 5-HT₇ receptors (Hachoumi et al., 2022) although this has not yet been demonstrated in mammals”.

Line 172: by 'increased hyperpolarisation' do you mean that the resting membrane potential was hyperpolarized? If so, this should be reworded for clarity.

The reviewer is correct. We have amended our terminology to indicate that 5-HT injected in the vicinity of the motoneurone caused hyperpolarisation of the resting membrane.

Line 174: while the work was conducted in JF Perrier's lab, I think it is important to acknowledge and the credit should be drawn to the first author, F Cotel, who likely performed the majority of the work. Please rephrase.

We have changed this sentence to read, “An underlying mechanism of 5-HT-mediated inhibition of motoneurone activity was clarified in a series of elegant experiments by Cotel and Perrier”.

Line 189: An interesting observation that I think is worth mention, that 5HT fibres have been reported at the AIS of rodent motoneurons (Deardorff, Romer, and Fyffe, 2021), which is in contrast to that has been suggested by Cotel et al. It is possible that this is a species difference. Alternatively, it might be due to differences in motoneuron types, which have not really been explored. Regardless, it still provides a means for direct compartmental modulation of channels that are expressed at the AIS, which differ from those expressed in the dendrites.

We agree with the referee that this is an important consideration. We have amended our text to read, “However, it is worth noting that these findings are only reported for adult turtle motoneurons and may not be applicable for other species. Indeed, a recent study have identified 5-HT boutons present on the AIS of rodent motoneurons (Deardorff et al., 2021), and compartmentalisation of 5-HT receptor subtypes on human motoneurons is yet to be detailed”.

Line 224: Indeed, previous work has demonstrated contributions of CaV and NaV channels to PIC and PIC-mediated firing properties, and historically it was thought that PICs were relatively straight forward. However more recent work from the Brocard group has suggested that it might be a bit more complex than initially thought as other ion channels can contribute to PIC and PIC-mediated intrinsic properties. Examples include activation of TRPM5 (Bos et al., 2021) and inactivation of KV1.2 (Bos et al., 2018). Further, M-type potassium currents (mediated by KCNQ channels) influence measures of PIC and their effect on intrinsic properties and oppose NaV1.6-mediated inward currents in excitatory interneurons. While this direct interaction has not been shown in motoneurons, it is worth noting given that motoneurons do express KCNQ channels in somatodendritic and AIS compartments alongside NaV1.6 channels (Verneuril et al., 2020). Whilst historically it was thought that PICs were straightforward, it is becoming more apparent that this is not the case as multiple channels contribute to and even oppose their actions in parallel. A similar argument could be made for modulation of 'PIC-channels' as changes in PIC-mediated intrinsic properties could be mediated through changes in some of these other channels. I highly recommend checking these works out and think that these points are worth highlighting in this review.

We have included the sentence below to highlight evidence of more complex mechanisms of PIC-like non-linearities in motoneurone behaviour.

“However, recent evidence shows that non-linearities in motoneurone discharge may have more complex mechanisms including activation of TRPm5 and inactivation of Kv1.2 channels (Bos et al, 2018, Bos et al, 2021)”.

Line 249: I would suggest softening this statement from PICs mediate serotonin's effect on motor performance, to PICs may partially contribute to serotonin's effect on motor performance. As previously highlighted, many other ion channels can be influenced by serotonin.

We agree that this statement was too strong, so we now suggest that “PICs may be partially responsible for 5-HT mediated increases in voluntary activation”.

Line 251: PICs and their influence on motoneuron intrinsic properties differ between motoneuron subtypes. This might be a good spot to highlight some of these works (Lee and Heckman 1998, Huh et al. 2017; Sharples and Miles 2021). I think it is also worth highlighting that motoneuron subtypes express different complements of ion channels and possibly (although not known) express different 5HT receptors or have inputs to different regions. Along these lines, studies of different muscles with varying muscle fibre

compositions (eg. Soleus, TA, etc) may also respond differently to neuromodulation. These could possibly contribute to task-differences in function and neuromodulatory control.

These comments have provided a valuable inclusion for our manuscript. We have amended our text to the following:

“These findings provide support that PIC activation in humans is linked to voluntary drive and hence 5-HT release. The difference between gastrocnemius and soleus also highlights that PICs, and their influence on motoneuron intrinsic properties, differ between muscles and motoneuron subtypes (Lee & Heckman, 1998; Huh et al., 2017)”.

Line 253-256: This statement isn't very clear. I suggest breaking it up a bit.

We have rewritten this statement to improve the clarity of the text:

“However, it is known that PICs associated with gastrocnemius medialis increase from 10% to 20% MVC, and PICs associated with soleus increase from 10% to 30% MVC, during slow ramped plantarflexions (Orssatto et al., 2021b). These findings provide support that PIC activation in humans is linked to voluntary drive and hence 5-HT release.”

Line 256: Not to mention that changes in the expression of the PIC channels or those that oppose their actions could also contribute.

We are not sure what the reviewer is suggesting here. Acute changes in Delta F and maximal voluntary force are seen after 5 muscle stretches of ~1 min each and then recover over ~10 min. We have not amended the manuscript based on this comment.

Line 313: suggest toning back language from '...in the spinal cord was due to...' to '...in the spinal cord could be due to...' as although plausible, this is purely speculative.

We have changed the phrase ‘in the spinal cord was due to’ to ‘in the spinal cord could be due to’ in the revised manuscript

Line 348: Modulated 'by' 5HT.

We have changed the phrase ‘modulated with 5-HT’ to ‘modulated by 5-HT’.

Line 369-373: Maybe not such a recent idea with the citations provided approaching 15 years ago. Suggest rephrasing.

We have changed the phrase ‘A recent proposal’ to ‘A reasonable proposal’.

Line 375: This paragraph ends kind of open. Is there a possibility to capitalise on new genetic tools in rodents that allow for identification and manipulation of defined interneuron subtypes to advance these ideas further? Are there new methods/technologies/techniques in humans that might provide some insight?

We agree that this paragraph finished quite open-endedly. However, we believe that the simple explanation for the open-ended paragraph was its lack of conclusion. In particular, the final sentence did not emphasize the take home message from the paragraph. We have changed the final sentence in this paragraph to read, “Thus, it is possible that a 5-HT mechanism involving inhibitory spinal circuitry can regulate the amplitude of agonist and antagonist muscle contractions”.

Line 377: I'm not entirely sure that this question was answered in this conclusion paragraph. Indeed, challenges are highlighted, but I think it would be worthwhile to outright statement that might provide more of an answer to this question.

We have changed this paragraph to provide more direction for the reader:

“Animal experiments provide clear evidence that 5-HT is a potent modulator of spinal circuits and motoneurone output. However, the effects of 5-HT on voluntary muscle activity in humans are less clear. Effects on motor performance during whole-body exercise are inconsistent. This is not to say that serotonergic neuromodulation does not matter for humans, but instead highlights the challenges associated with studying how a complex neuromodulatory system acts during muscle contractions. Controlled experiments using single-joint, single muscle, contraction protocols have found that maximal force can be changed by altering 5-HT activity in the CNS. The differences observed in voluntary activation are small but are present despite an otherwise intact system and the actions of other neuromodulators, including noradrenaline. This suggests that serotonin has a non-redundant role in maximal voluntary contractions but still begs the question of exactly how important it is in typical motor tasks. Nonetheless, 5-HT-related changes in muscle activation typically emerge with strong contractions for both the unfatigued and fatigued motor system. Thus, it appears that the magnitude of descending drive to the muscle may be aligned with the level of 5-HT neuromodulation in humans. Indeed, we are beginning to reveal evidence where 5-HT-effects may be scaled to the intensity of muscle activation in humans (Goodlich *et al.*, 2022; Henderson *et al.*, 2022)”.

Line 378: neuromodulation of what? This sentence is rather vague as 5HT is a neuromodulator and it can be inferred that it would contribute to neuromodulation. I would suggest stating that 5HT is a potent modulator of spinal circuits and motoneuron output.

As suggested, we have changed this sentence to, “Animal experiments provide clear evidence that 5-HT is a potent modulator of spinal circuits and motoneurone output”.

Line 393-396: this sentence is a little misleading as it implies that pharmacological manipulation of neurotransmitters in the CNS is unique to humans. However, I think that the main point is detailed in the sentence that follows. Perhaps consider rephrasing these two sentences.

This was a poorly worded sentence where we should not have used the word 'unique'. We have removed this term and stated in the revised manuscript:

Participant safety is an additional challenge in human experiments that use pharmacology to manipulate neurotransmitter activity. Human studies must operate within a window of safe drug administration, and typically use therapeutic doses of 5-HT modulating medications. Thus, very little is known about how dosage effects influence 5-HT activity in humans.

Figure 1: I think this figure is a little problematic as it attempts to synthesize data from multiple species (turtle and mammals). It should be made clearer in the figure itself (in addition to caption) where these data are derived otherwise it is a bit misleading. As highlighted above, it should be highlighted that the DLF contains descending serotonergic fibres and is not simply an analogue of the raphe spinal pathway.

We thank Referee #3 for this suggestion. Please note that the intent of the figure was to provide an overview of the 5-HT system and not to provide specific details across species. If we were to modify the figure itself to highlight where all of the data was derived from, it would compromise the layout and clarity of its content. Instead, we have highlighted for caption 1A that the subsequent information is derived from the mammalian spinal cord, and for caption 1B that the subsequent information is derived from preparations of the adult turtle spinal cord. The source reference for the turtle study is listed at the bottom of the caption.

REFeree #2

It was with pleasure that I read the work submitted by Kavanagh and Taylor entitled "Voluntary activation of muscle in humans: does serotonergic neuromodulation matter?" For many years, the field has been left wondering if PICs that are facilitated by 5HT are actually relevant to human function. This topical review starts to address this, and other, aspects of 5HT neuromodulation during normal and fatigued motor output. It is with great certainty that I can say that this article will be highly read and cited. Great work! Below, I have provided some specific comments (with page ; and line [l]) in an attempt to improve the quality of the manuscript.

We thank Referee #2 for these kind comments. We believe that our Topical Review has significantly improved with the recommendations that the referee has made.

p2; l57: Iontropic/neuromodulatory is not necessarily a class of input, rather they are inputs that activate classes of receptors. The inputs activate either ionotropic or metabotropic receptors and neuromodulatory inputs would predominantly activate those of the metabotropic class. Slight rewording for clarity would be helpful here.

We have modified this sentence to read, "Synaptic inputs to motoneurons will typically activate two classes of receptors: ionotropic and neuromodulatory".

p5; l137: is this supposed to refer to rates of discharge?

The referee is correct. We have modified the sentence to clarify that we are referring to discharge rate, “Incremental increases in treadmill walking speed correspond to incremental increases in the discharge rate of single raphe-spinal fibres”.

Section ending on l142: The discussion of the role of 5ht on the dorsal cord seems a little light - one might consider a slight expansion of this topic here so that the (predominantly) inhibitory effects of 5ht on sensory transmission can be appreciated.

We have added additional information to this paragraph. In doing so, we have been cautious with the amount of information that we present. In particular, the role that 5-HT plays in regulating afferent and interneuron excitability is extraordinarily complex, and to do justice to the topic would require a review by itself. We have included the following information in the revised manuscript:

“5-HT release into the dorsal horn and intermediate zone of the spinal cord can cause remarkably complex outcomes at the neuronal level, as 5-HT can either depress or facilitate transmission in afferent fibres (Belcher et al., 1978; Jordan et al., 1979; Todd & Millar, 1983), and the response to 5-HT can differ depending on the threshold for afferent activation (Belcher et al., 1978; Jankowska et al., 1993). Adding further complexity to our understanding of how 5-HT regulates sensory neurone in the spinal cord is that 5-HT effects may be exerted pre-synaptically and post-synaptically. In particular, actions of muscle spindle (Ia fibres) and tendon organ (II fibres) afferents on spinal interneurons by 5-HT is dependent on the type of the afferent that is activated and the functional type of the interneuron it is connected to (Jankowska et al., 2000). Thus, it is difficult to predict how motor function may be influenced by the actions of 5-HT on afferent neurons and interneurons”.

p7; l207-209: It may be nice to put this magnitude of change into perspective for the naive reader. If a healthy young adult has near complete activation of the biceps (near 100%; see pre-fatigue %VA in Fig 3) then we would not expect that they could gain more than a percent or two as they cannot have >complete activation (theoretical, I know). Other muscles, with lower %VA, may show greater change.

We agree that more detail could have been provided to give our voluntary activation data context for a naïve audience. Hence, we have modified the text as follows:

A finding of only small drug-related increases in voluntary activation is perhaps not surprising, as increases in motoneurone excitation during near-maximal contraction intensities produce only small changes in interpolated twitch amplitude (Herbert & Gandevia, 1999). Hence, voluntary activation of the biceps (that is calculated from interpolated twitches) may be as high as 98% or 99% in healthy individuals, and any intervention that may increase activation is limited by a very close ceiling (i.e. 100% voluntary activation).

p8; l235: put "i.e. the motoneurone's hysteresis" in parenthesis?

We have placed these words in parenthesis in the revised manuscript.

p8; 1236-237: Although seldomly considered in the literature, the threshold of the PIC can vary and may be a key contributor to MN recruitment. Please refer to the discussion of Afsharipour et al 2021

This is an excellent suggestion and we have modified the text to read, “PICs provide little drive to the motoneurone at recruitment as initiation of an action potential and the activation of PICs occur at a similar membrane potential (but see Afsharipour et al 2020 for important nuances with regard to variations in PIC threshold).”

Section ending l375: I suggest updating this section with reference to the the emerging findings from both CJ Heckman's lab and from the second author's lab (references below). These findings, albeit preliminary in nature, both support the notion that afferent feedback may dampen the effects of PICs in humans: Pearcey et al. 2020 - Exploring the effects of Ia reciprocal inhibition on neuromodulatory commands in the human lower limb (<https://doi.org/10.1096/fasebj.2020.34.s1.09445>); Mesquita et al. 2022 - Effects of reciprocal inhibition and whole-body relaxation on persistent inward current estimated by two different methods (<https://doi.org/10.1113/JP282765>)

We have included the following text in this section that highlights the findings of Pearcey et al and Mesquita et al.:

“The notion that reciprocal inhibition can dampen the effects of PICs has also been demonstrated in humans, where 128 Hz vibration of the tibialis anterior tendon to activate dorsiflexor muscle spindle afferents decreases ΔF for both the soleus and medial gastrocnemius during 30% MVC plantarflexions, and vibration of the Achilles tendon decreases ΔF for the tibialis anterior during 30% MVC dorsiflexions (Pearcey et al., 2020). Another human study found that 1 Hz electrical stimulation of the common peroneal nerve also has the capacity to reduce ΔF in medial gastrocnemius motor units in healthy individuals, which builds further support that Ia reciprocal inhibition reduces the contribution of PICs to MU firing in humans. (Mesquita et al., 2022)”.

REFeree #3

This review provides a comprehensive and balanced overview of the role of 5-HT on muscle activation. Insights from mammalian and from human studies are discussed. The authors do a commendable job of comparing the results from both animal and human studies and identify methodological challenges in testing hypotheses derived from animal work in human studies. The review will be very appreciated by the motor control community. I only have minor comments:

We thank Referee #3 for reviewing our manuscript and providing positive comments. We have carefully considered each comment that has been provided and we have amended the manuscript accordingly.

1. I know that this review looks at voluntary muscle activation but I wonder if there are any insights about 5-HT perhaps modulating reflexive movements in humans and

whether there could be a greater role in that type of movement based on the possibility that serotonergic systems are activated by novel or surprising stimuli.

Given that the focus of this Topical Review is voluntary muscle activation we are cautious to present additional data that would not align with our theme. However, we believe that amendments we have made to the manuscript so far may address Referee #3's suggestion. In particular, the third paragraph of the section The serotonergic system now reads:

5-HT release into the dorsal horn and intermediate zone of the spinal cord can cause remarkably complex outcomes at the neuronal level, as 5-HT can either depress or facilitate transmission in afferent fibres (Belcher et al., 1978; Jordan et al., 1979; Todd & Millar, 1983), and the response to 5-HT can differ depending on the threshold for afferent activation (Belcher et al., 1978; Jankowska et al., 1993).

Furthermore, our amendment to the section titled Afferent feedback can potentially regulate 5-HT effects at the motoneurone now includes (amongst other things):

The likely candidate for this afferent mechanism is Ia disynaptic reciprocal inhibition evoked by length changes in the antagonist muscle, as other afferents are relatively insensitive to the changes in muscle length that occurred with passive rotation of the cat ankle in the experiment (Hynstrom *et al.*, 2007). The notion that reciprocal inhibition can dampen the effects of PICs has also been demonstrated in humans, where 128 Hz vibration of the tibialis anterior tendon to activate dorsiflexor muscle spindle afferents decreases ΔF for both the soleus and medial gastrocnemius during 30% MVC plantarflexions, and vibration of the Achilles tendon decreases ΔF for the tibialis anterior during 30% MVC dorsiflexions (Pearcey *et al.*, 2020). Another human study found that 1 Hz electrical stimulation of the common peroneal nerve also has the capacity to reduce ΔF in medial gastrocnemius motor units in healthy individuals, which builds further support that Ia reciprocal inhibition reduces the contribution of PICs to MU firing in humans. (Mesquita *et al.*, 2022).

2. What about the role of 5-HT in mediating presynaptic inhibition of sensory afferents? Are there insights on this in humans, and does it relate to central fatigue?

Once again, we believe that our amendment to the section The serotonergic system may address this comment. Overall, the role that 5-HT plays on regulating excitability in the sensory system is complex, and the role that 5-HT has in mediating presynaptic inhibition of sensory afferents warrants a review by itself. However, we have briefly described this section:

“Adding further complexity to our understanding of how 5-HT regulates sensory neurone in the spinal cord is the 5-HT effects that may be exerted pre-synaptically and post-synaptically. In particular, actions of muscle spindle (Ia fibres) and tendon organ (II fibres) afferents on spinal interneurons by 5-HT is dependent on the type of the afferent that is activated and the functional type of the interneuron it is connected to (Jankowska et al., 2000). Thus, it is difficult to predict how motor function may be influenced by the actions of 5-HT on afferent neurons and interneurons. Instead, the most direct effects of 5-HT on motor function are via the monosynaptic connections to the motoneurons from the fibres in the raphe-spinal pathway”.

3. If I can make a suggestion to add Bui et al. (2003) (work from Ken Rose lab) to the works cited in line 356.

We do not believe that this reference can be incorporated into the text that the referee has drawn our attention to. Our statement indicates that reciprocal inhibition may be especially critical for regulating PIC amplitude during functional motor activity, where even minor rotations of the ankle joint will reduce PICs by ~50%. The experiments in Bui et al., show that spinal motoneurons, Ia inhibitory interneurons, and Renshaw cells differ in their ability to deliver current from dendritic synapses to the soma and to transmit voltage changes along their dendrites. The work by Bui et al, examines dendritic geometry without providing insight to PICs or reciprocal inhibition.

Dear Dr Kavanagh,

Re: JP-TR-2022-282565R1 "Voluntary activation of muscle in humans: does serotonergic neuromodulation matter?" by Justin J Kavanagh and Janet L Taylor

I am pleased to tell you that your Topical Review article has been accepted for publication in The Journal of Physiology, subject to any modifications to the text that may be required by the Journal Office to conform to House rules.

NEW POLICY: In order to improve the transparency of its peer review process The Journal of Physiology publishes online as supporting information the peer review history of all articles accepted for publication. Readers will have access to decision letters, including all Editors' comments and referee reports, for each version of the manuscript and any author responses to peer review comments. Referees can decide whether or not they wish to be named on the peer review history document.

The last Word version of the paper submitted will be used by the Production Editors to prepare your proof. When this is ready you will receive an email containing a link to Wiley's Online Proofing System. The proof should be checked and corrected as quickly as possible.

All queries at proof stage should be sent to tjp@wiley.com

The accepted version of the manuscript will be published online, prior to copy editing in the Accepted Articles section.

Are you on Twitter? Once your paper is online, why not share your achievement with your followers. Please tag The Journal (@jphysiol) in any tweets and we will share your accepted paper with our 22,000+ followers!

Yours sincerely,

Professor Laura Bennet
Senior Editor
The Journal of Physiology
<https://jp.msubmit.net>
<http://jp.physoc.org>
The Physiological Society
Hodgkin Huxley House
30 Farringdon Lane
London, EC1R 3AW
UK
<http://www.physoc.org>
<http://journals.physoc.org>

*** IMPORTANT NOTICE ABOUT OPEN ACCESS ***

To assist authors whose funding agencies mandate public access to published research findings sooner than 12 months after publication The Journal of Physiology allows authors to pay an open access (OA) fee to have their papers made freely available immediately on publication.

You will receive an email from Wiley with details on how to register or log-in to Wiley Authors Services where you will be able to place an OnlineOpen order.

You can check if your funder or institution has a Wiley Open Access Account here <https://authorservices.wiley.com/author-resources/Journal-Authors/licensing-and-open-access/open-access/author-compliance-tool.html>

Your article will be made Open Access upon publication, or as soon as payment is received.

If you wish to put your paper on an OA website such as PMC or UKPMC or your institutional repository within 12 months of publication you must pay the open access fee, which covers the cost of publication.

OnlineOpen articles are deposited in PubMed Central (PMC) and PMC mirror sites. Authors of OnlineOpen articles are permitted to post the final, published PDF of their article on a website, institutional repository, or other free public server, immediately on publication.

Note to NIH-funded authors: The Journal of Physiology is published on PMC 12 months after publication, NIH-funded authors DO NOT NEED to pay to publish and DO NOT NEED to post their accepted papers on PMC.

EDITOR COMMENTS

Reviewing Editor:

The authors have done a great job in incorporating the reviewer's suggestions in their revised manuscript. All reviewers are satisfied and enthusiastic about this review submission.

REFEREE COMMENTS

Referee #1:

The authors have done an excellent job at addressing my concerns. The manuscript is stronger as a result and I anticipate will be highly cited within the field.

Referee #2:

The authors have done an excellent good job improving their manuscript based on the comments provided from reviewers. Great work!

Referee #3:

The authors have addressed my suggestion appropriately. Apologies for the last suggestion I made. I meant to point the authors towards Bui et al. 2008, not 2003. But it really is a very minor comment, and the revised manuscript is fine.

1st Confidential Review

21-Jun-2022